# Epithelial plasticity and innate immune activation promote lung tissue remodeling following respiratory viral infection

Andrew K. Beppu[1,2,3,5], Juanjuan Zhao[1,2,3,5], Changfu Yao [1,2,3], Gianni Carraro [1,2,3], Edo Israely[3], Anna Lucia Coelho[1,2], Katherine Drake [1,2,3], Cory M. Hogaboam[1,2], William C. Parks[1,2], Jay K. Kolls [4] & Barry R. Stripp [1,2,3] ✉

Epithelial plasticity has been suggested in lungs of mice following genetic depletion of stem cells but is of unknown physiological relevance. Viral infection and chronic lung disease share similar pathological features of stem cell loss in alveoli, basal cell (BC) hyperplasia in small airways, and innate immune activation, that contribute to epithelial remodeling and loss of lung function. We show that a subset of distal airway secretory cells, intralobar serous (IS) cells, are activated to assume BC fates following influenza virus infection. Injury-induced hyperplastic BC (hBC) differ from pre-existing BC by high expression of IL-22Ra1 and undergo IL-22-dependent expansion for colonization of injured alveoli. Resolution of virus-elicited inflammation results in BC to IS re-differentiation in repopulated alveoli, and increased local expression of protective antimicrobial factors, but fails to restore normal alveolar epithelium responsible for gas exchange.

Stem cell plasticity contributes to tissue regeneration and includes a range of processes such as lineage conversion and de-differentiation of specialized progenitor cells[1]. Although epithelial cell plasticity in lung repair has been inferred from lineage tracing and targeted cell ablation studies[2,3], the physiological relevance in lung disease is unknown. The epithelial linings of airways and alveoli are maintained by distinct regional facultative stem/progenitor cells whose progeny include both self-renewing and differentiating subsets[4,5]. Regional differences in the fate of differentiating progeny allow for the maintenance of locally specialized epithelial functions, such as mucociliary clearance and host defense in the conducting airways and gas exchange in the distal respiratory units. Restriction of progenitor cells and their differentiating progeny to distinct anatomic zones during homeostatic tissue maintenance are necessary for the functional integration of these compartments along the proximodistal axis and preservation of normal physiological lung function. However, acute lung injury and chronic lung disease disrupt normal progenitor cell compartmentalization leading to aberrant tissue remodeling and defective alveolar regeneration[6,7]. Such is the case following infections by zoonotic respiratory viruses such as H1N1 influenza A or SARS-CoV2, and in chronic lung diseases such as idiopathic pulmonary fibrosis (IPF). In such conditions, basal cell (BC) hyperplasia in airways leads to the recruitment and colonization of these injured alveolar areas, and this proximalization of distal lung tissue contributes to potentially life-threatening loss of alveolar diffusion capacity[7–12]. However, the identity of epithelial progenitor cells that contribute to the proximalization of distal lung tissue and the mechanisms that regulate their fate during tissue remodeling remain poorly defined.

Mirroring what occurs in injured or infected human lungs, acute lung injury in mice infected with a mouse-adapted Puerto Rico 8 (PR8) variant of the H1N1 influenza virus is accompanied by BC expansion in airways that ultimately replaces the injured epithelium of the alveolar

[1]Department of Medicine, Women's Guild Lung Institute, Cedars-Sinai Medical Center, Los Angeles, CA 90048, USA. [2]Division of Pulmonary and Critical Care Medicine, Department of Medicine, Cedars-Sinai Medical Center, Los Angeles, CA 90048, USA. [3]Department of Medicine, Regenerative Medicine Institute, Cedars-Sinai Medical Center, Los Angeles, CA 90048, USA. [4]Tulane Center for Translational Research in Infection and Inflammation, School of Medicine, New Orleans, LA 70112, USA. [5]These authors contributed equally: Andrew K. Beppu, Juanjuan Zhao. ✉e-mail: barry.stripp@csmc.edu

gas-exchange region[8,9,13]. Even though basal and club cells serve as stem cells for maintenance of the pseudostratified epithelium of proximal airways and cuboidal epithelium of distal airways, respectively, lineage tracing studies suggest that hyperplastic BC appearing in distal lung tissue of PR8-infected mice are derived from neither of these canonical stem cell populations[8,9,14]. Instead, injury-induced hyperplastic BC (hereon referred to as hBC) derives from alternate small airway progenitors that can be lineage traced based upon expression of either Sox2 or p63 transcription factors[14–16]. Similarly, in proximal airways, α-smooth muscle actin-expressing myoepithelial cells of submucosal glands (SMG) or SSEA4-expressing secretory cell progeny of BC can replenish basal stem cells following injury[2,17]. However, the molecular and functional relationship between epithelial progenitors of SMG or upper airway surface epithelium that can replace local basal stem cells, versus those that yield expanding BC in small airways following PR8 infection, remains to be established.

Although much is known about mechanisms that regulate the renewal and fate of BC within pseudostratified airways, mechanisms regulating hyperplastic BC appearing in the alveolar epithelium of PR8-infected mice are poorly defined. Evidence that localized lung hypoxia[15] and the altered inflammatory milieu[18–20] that accompanies PR8 infection may regulate tissue remodeling suggests potential roles for innate immune activation as a regulator of BC fate and altered epithelial progenitor cell function. Epithelial-immune crosstalk serves as a critical regulator of progenitor cell function in multiple organs including the lung, gut, and skin[21–25].

Here, we show that the unique origins and molecular phenotype of hBC elicited by PR8 infection allow for their dynamic response to the activated innate immune system. Interleukin 22 derived from locally activated γδT cells promotes self-renewal and hyperplasia of BC within alveolar epithelium that ultimately assumes serous cell fates in previously injured regions during resolution of the PR8-elicited inflammatory response. This remodeling response to respiratory viral infection allows for efficient replacement of exposed basement membrane in the injured alveolar epithelium and local production of an antibacterial secretome to protect against secondary bacterial infection.

## Results

### Activation and expansion of intralobular serous (IS) cells precedes BC hyperplasia following influenza-induced acute lung injury

BC hyperplasia occurs in the airways of patients with chronic lung disease and in the lung tissue of those succumbing to acute respiratory viral infections such as the H1N1 influenza virus or COVID-19[8,9,26–28]. Based on lineage tracing studies, the majority of cells in mouse lungs that acquire BC fates in response to influenza virus-induced acute lung injury have been proposed to arise from immature P63+Krt5- basal progenitors[15,16]. However, a significant fraction of hBC seen during injury failed to retain a P63 lineage tag suggesting the presence of other contributing non-basal progenitors[29]. We sought to interrogate the existence of non-canonical progenitors by assessing dynamic changes in epithelial cell populations in response to influenza-induced acute lung injury. We generated a comprehensive profile of single-cell transcriptomes for epithelial cells isolated from the trachea, extrapulmonary bronchus, intralobular airways, and alveolar regions of control C57Bl/6 mice (Day 0) and of mice infected and recovering from exposure to the PR8 strain of H1N1 influenza virus (Day 3–240; Fig. 1A, B). Results displayed as uniform manifold approximation and projection (UMAP) two-dimensional plots reveal eight major cell clusters that were categorized according to known lung epithelial cell types based on their unique gene expression signatures (Fig. 1B, C; Supplementary Fig. 1A). Even though each of the major cell types was observed among epithelial cells sampled from all control and post-exposure time points, their relative proportions within total sampled epithelial cells showed significant variability (Supplementary Fig. 1B

and C). Basal cells were only infrequently observed among epithelial cells sampled from lobes of control mice but showed progressive increases in representation at time points evaluated following PR8 infection (Fig. 1D, E). Serous cells were the only other rare epithelial cell type in the lung lobes of naïve mice whose abundance increased following PR8 infection (Fig. 1D, E). However, the increase in representation of serous cells preceded increases in BC representation among epithelial cells sampled across the time course (Fig. 1E).

Because serous cells show increased representation in single-cell data sets at earlier stages in the response to PR8 infection, we speculated that they may represent the progenitor cell type of origin for hBC. To further explore this concept, we interrogated our scRNAseq data to identify differentially expressed genes (DEGs) that discriminate serous cells from other epithelial cell types of conducting airways. Both pre-existing serous cells of steady-state airways and expanding populations of serous cells observed during the early injury/inflammatory phase following PR8 infection were unique among distal lung epithelial cell types in their expression of genes such as *Ifitm1*, *Ifitm3*, *Bpifa1*, and *Ltf*, that were associated with anti-microbial host defense (Fig. 1F; Supplementary Figs. 1D, 2A). Interferon-regulated genes such as Ifitm1 and Ifitm3[30] were notably induced among serous cells during the early injury/inflammatory phase of PR8 infection, thus presumably providing a survival advantage to this pool of airway progenitors. Similarly, other antimicrobial genes such as *Bpifa1* and *Ltf*, whose expression defines serous cells of proximal airways and submucosal glands[17], were both induced following PR8 infection and served to distinguish the transcriptomes of serous cells from the other bronchiolar secretory cell type, club cells (Fig. 1F; Supplementary Fig. 2A). We found that both serous and club cells express high levels of the secretoglobin *Scgb3a2*, but that club cells were unique in their expression of another secretoglobin family member, *Scgb1a1*. Furthermore, serous cells of the PR8-exposed lung include a subset that harbor transcripts for BC marker genes including *Trp63*, *Krt14*, and *Krt5*, present at variable levels among all post-exposure recovery time points but absent in airways of naïve mice (Fig. 1G). Immunofluorescence staining revealed that a small fraction of intralobular Scgb3a2-immunoreactive cells were also positive for Msln and Bpifa1, indicating the presence of serous cells within the lower respiratory tract, hereon called intralobular serous (IS) cells (Fig. 1H). Expansion of IS cells coupled with the acquisition of a transcriptome that shares similarities with nascent airway BC 14 days following PR8 exposure, provided indirect support for the notion that airway serous cells represent the progenitor cell-of-origin for hBC observed in airways and alveolar epithelium of PR8-infected mice.

### Transcriptome analysis predicts IS>basal plasticity during recovery from PR8 infection

To further examine the potential for IS cells to serve as progenitors for the expansion of hBC, we used gene set enrichment to identify DEGs among these epithelial cell clusters at different times post-PR8 infection. We observed a significant induction of gene sets associated with cell cycle at 5 and 7 days post-infection coinciding with the appearance of proliferating conducting airway epithelium observed in vivo (Fig. 2A). However, we observed enrichment of cell-cycle gene sets at these time points only in serous and BC with little to no enrichment of these gene sets in other major cell types of conducting airways (Fig. 2A). We used our scRNAseq dataset to interrogate lineage relationships between BC, IS cells, and other airway secretory cell populations. Transcript splicing was assessed by Velocyto[31] to predict differentiation trajectories among IS cells, club cells, and hBC during early (naïve-day 14) and late (day 14-day 21) responses to PR8 infection (Fig. 2B, C). Computational analysis of cell fate dynamics predicted a process that IS cells de-differentiated into BC at early time points post-PR8 infection (Fig. 2C). In contrast, trajectory analysis predicted the transition of BC back to IS cells at late recovery time points (Fig. 2C).

 

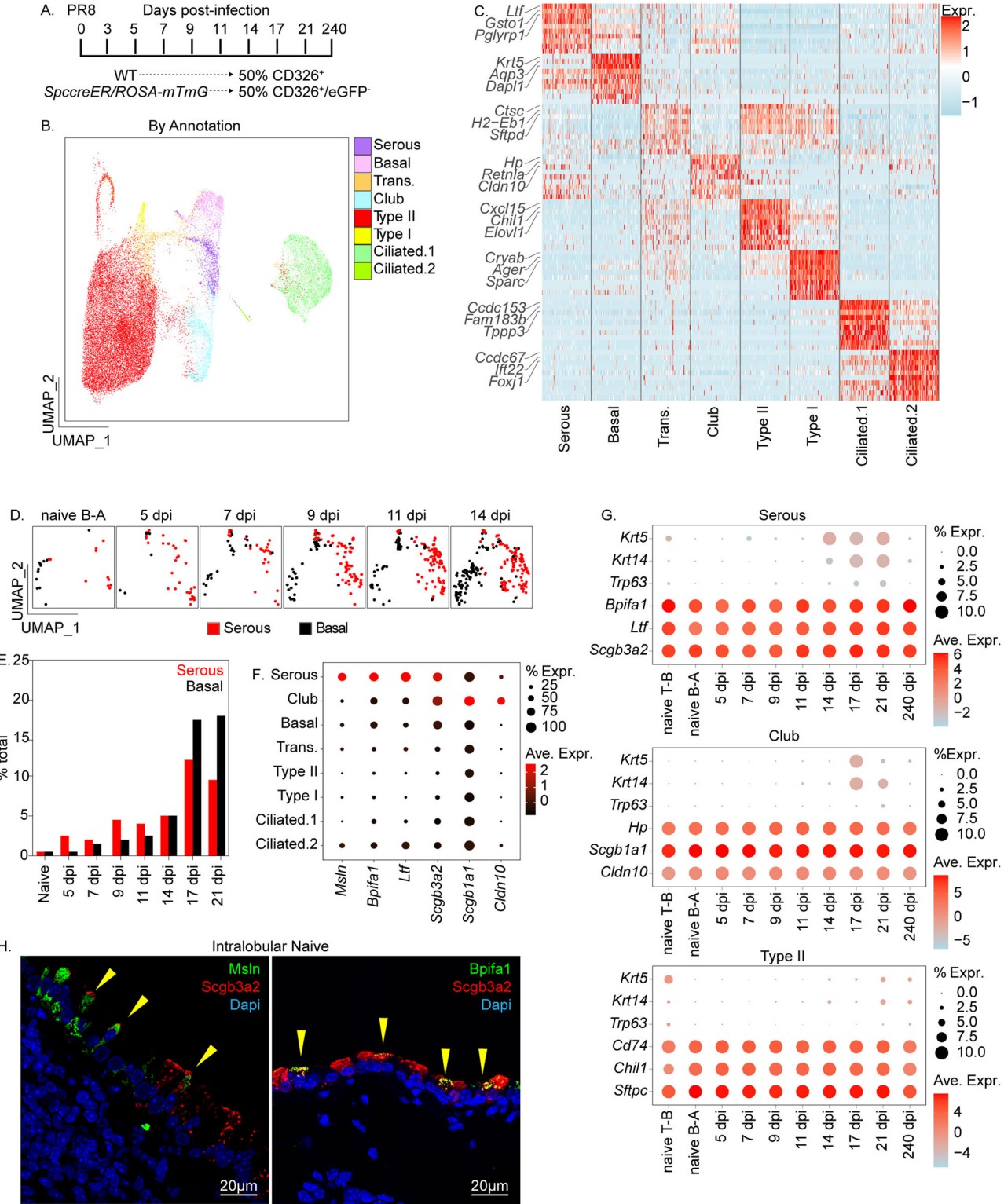

We next used Visium spatial RNA-Seq to define the relationship between hBC and IS cells within lung tissue 14 days after PR8 infection (Fig. 2D). Unsupervised clustering revealed 7 transcriptionally distinct gene signatures that were projected onto a spatial map of the sampled tissue section (Fig. 2E). Cell type signature scores generated from scRNAseq (Fig. 1C) were applied to spatial RNAseq data to colocalize cell types (Fig. 2F). The resulting spatial maps of cell types demonstrated close spatial association of IS but not club cells with hBC's, as suggested from the predicted lineage relationships between these cell types by RNA velocity analysis (Fig. 2F, G; Supplementary Fig. 2B). We corroborated spatial RNAseq data by immunofluorescent localization of Scgb3a2 to regions of Krt5-immunoreactivity within remodeled regions of the alveolar epithelium of PR8-infected mice (Fig. 2H).

## Scgb3a2$^{pos}$Scgb1a1$^{neg}$ IS cells demonstrate serous-basal-serous plasticity during repair following PR8

To validate our earlier predictions that IS but not club cells could serve as precursors for hBC in response to PR8 infection, we developed a

**Fig. 1 | ScRNAseq to reveal the molecular phenotype of lung epithelial cells during the course of PR8 influenza virus infection. A** Experimental design. Mouse lung homogenates were collected from 8 to 12-week-old C57/Bl6 and *Sftpc-CreER/ROSA-mTmG* mice (*n* = 5 per group). Lung tissue was collected at indicated time points and either total epithelial cells (C57/Bl6 mice; CD31⁻CD45⁻CD326⁺) or AT2-depleted epithelial cells (*Sftpc-CreER/ROSA-mTmG* mice; CD31⁻CD45⁻CD326⁺eGFP⁻) enriched by FACS. **B** UMAP plot of combined scRNAseq data. Unsupervised clustering was used to distinguish distinct cell phenotypes which were assigned to known epithelial cell types based upon gene signatures. Pie chart shows a fractional representation of each cell types within the entire data set. **C** Heatmap showing unique molecular profiles between cell types. Top 3 genes for each cell category are annotated to the left. Full gene lists for each cell category are provided in

Supplementary Fig. 1A. **D** UMAP plot tracking relative changes in basal and serous cell populations for indicated samples. B-A bronchoalveolar. **E** Representation of serous (red bars) and basal (black bars) cell types as a function of percent total sampled cells at each time point. Analysis was performed on AT2-depleted samples (i.e. using the FACs enrichment strategy CD31⁻CD45⁻CD326⁺eGFP⁻ from *Sftpc-CreER/ROSA-mTmG* mice). dpi days post-infection. **F** Dot plot comparing expression of selected club and serous cell-specific genes between cell types. **G** Assessment of BC gene signature from PR8-induced injury. T-B tracheal bronchial. **H** Representative immunofluorescence colocalization of either Msln or Bpifa1 (green) with Scgb3a2 (red) within conducting airway epithelium (1 experiment, *n* = 5 biological replicates).

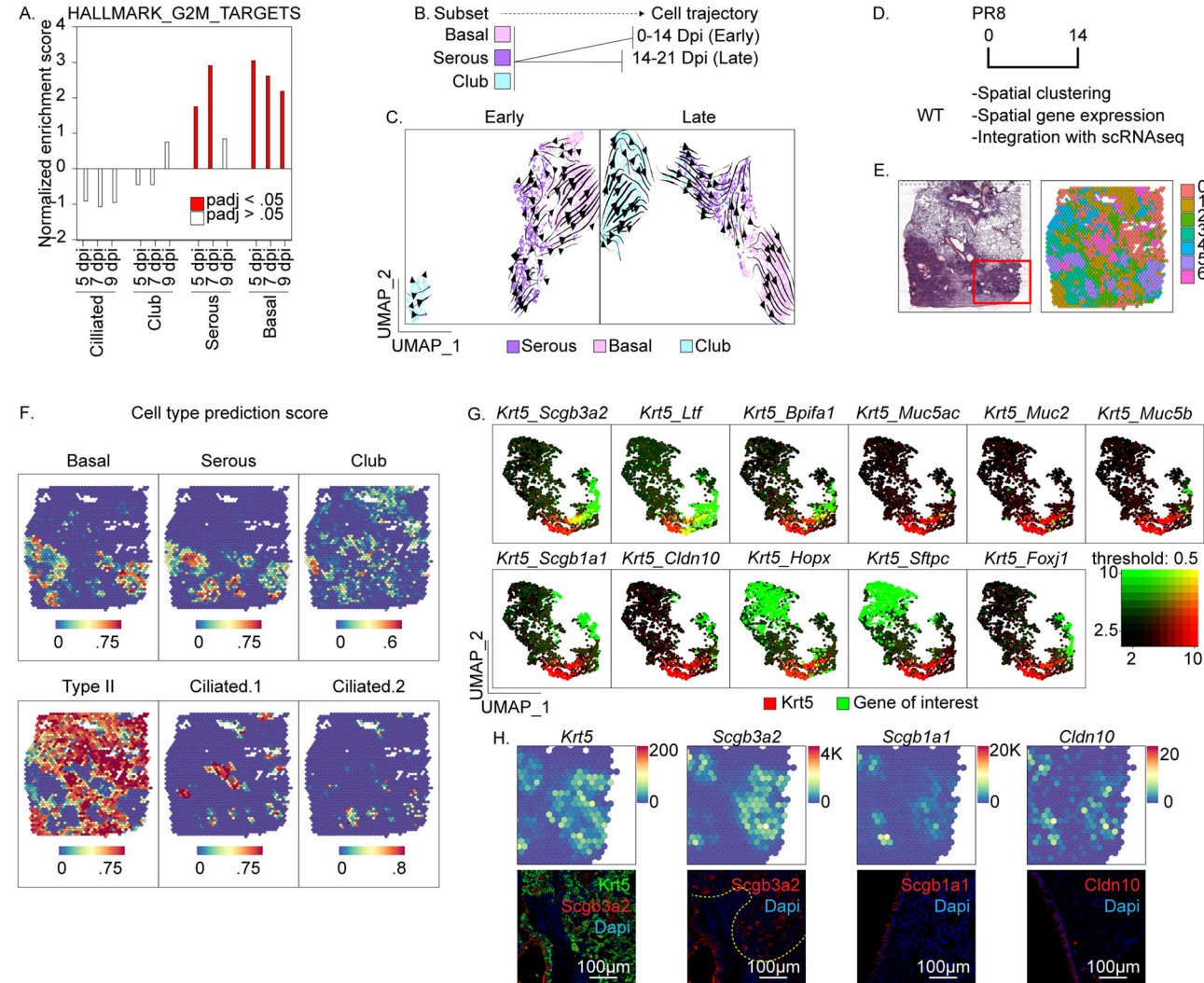

**Fig. 2 | Single-cell and spatial gene expression profiling to investigate lineage relationships between serous and basal cells following PR8 infection.**
**A** Comparative analysis of normalized enrichment for mitotic gene sets by cell type & condition. Red bars represent conditions with adjusted *p*-value < 0.05. *p*-value calculation is based upon an adaptive multi-level split Monte-Carlo scheme.
**B** Experimental design for the generation of RNA velocity projections. Single-cell RNAseq data were first subsetted by cell type (basal, serous, and club), followed by time following infection (early: naïve, 3, 5, 7, 9, 11, 14 dpi, or late: 14, 17, and 21 dpi). **C** Trajectory inference based upon RNA velocity profiling among basal, serous, and club cell subsets. **D** Experimental design for the generation of spatial gene expression data. **E** Projection of spot clusters onto H&E image of the tissue sample.
**F** Cell type prediction scores of epithelial populations represented within spatial

RNAseq data. Cell type-specific gene signatures were generated using scRNAseq data in Fig. 1A. Color scale at the bottom reflects the intensity of cell type prediction scores. **G** Co-expression of selected epithelial markers 14 days following PR8 exposure: *Krt5* (basal), *Scgb3a2* (serous and club), *Ltf, Bpifa1* (serous*), Muc5ac, Muc2, Muc2* (goblet), *Scgb1a1, Cldn10* (club), *Hopx* (Type I), *Sftpc* (Type II), *Foxj1* (Ciliated). **H** Spatial gene expression and corresponding immunofluorescence of cell type-specific markers. Transcripts were mapped onto spot coordinates from the region sampled in the red box in Fig. 2E: *Krt5* (basal), *Scgb3a2* (serous and club), *Scgb1a1* (club), and *Cldn10* (club) markers. Immunofluorescence colocalization of either Cldn10, Scgb1a1, or Scgb3a2 (red), with Krt5 (green), within the injured distal airway of 14 dpi PR8 infected mice (1 experiment, *n* = 5 biological replicates). Color scale reflects the abundance of indicated transcripts.

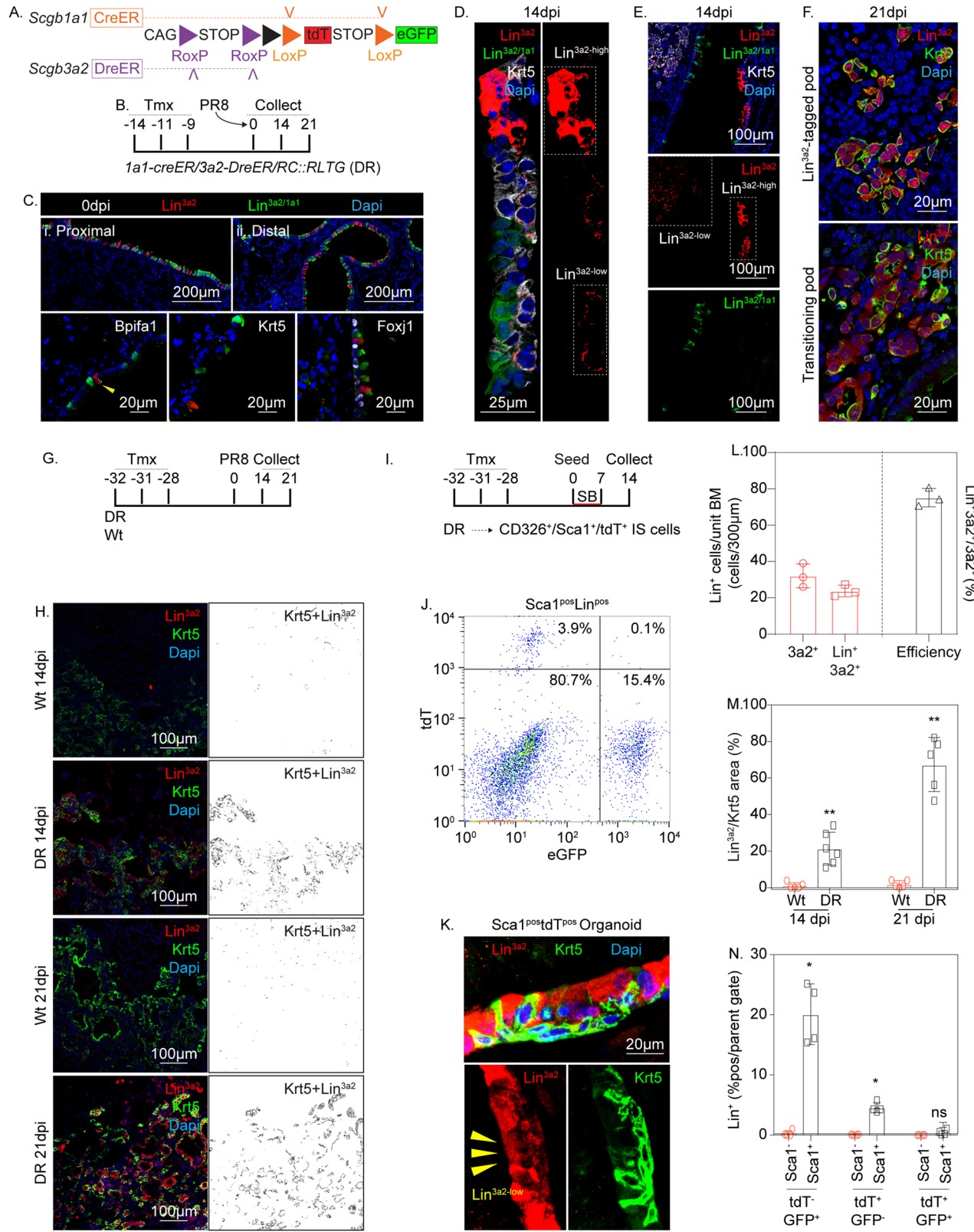

lineage tracing approach to independently tag and fate map these secretory cell types. Our single-cell and spatial RNAseq data indicated that both IS cells and club cells express Scgb3a2 but that only club cells express Scgb1a1. Accordingly, we generated dual recombinase (DR) mice harboring a Dre/Cre reporter allele in conjunction with *Scgb1a1-CreER* and *Scgb3a2-DreER* recombinase driver alleles (Fig. 3A). Tamoxifen exposure of these mice resulted in Dre-mediated excision

of a poly A signal (STOP) from the *RC::RLTG* dual recombinase reporter allele (Dre/Cre recombinase reporter; Fig. 3A) within Scgb3a2-expressing IS and club cells, with subsequent Cre-mediated excision of tdT-STOP within Scgb1a1-expressing club cells. Outcomes of these recombination events include tracing of Scgb3a2+/Scgb1a1- IS cells by expression of tdT (Lin3a2), and Scgb3a2+/Scgb1a1+ club cells by expression of eGFP (Lin3a2/1a1; Fig. 3A). For lineage tracing after PR8

**Fig. 3 | Rare Scgb3a2$^+$/Scgb1a1$^-$ serous cells reside within intralobar epithelium and assume BC fates in response to PR8-induced airway injury. A** Schematic illustration of recombinase driver and reporter alleles used in Dre/Cre recombinase (DR) mice for independent lineage labeling of serous and club cells. **B** Experimental design. DR mice (*n* = 5 per experimental group) were treated with three doses of TM, infected with PR8, and recovered for the indicated time points. **C** Representative immunofluorescence localization of tdT (Lin$^{3a2}$; red) and eGFP (Lin$^{3a2/1a1}$; green) lineage reporters reflective of serous and club cells, respectively (1 experiment, *n* = 3 biological replicates). Shown is a low-magnification image (left) and selected high-magnification images of either proximal (i) or distal (ii) airway epithelium (right). Bottom row shows representative immunofluorescence colocalization of lineage reporters (green or red) with either Bpifa1, Foxj1, or Krt5 (white) in the airways of naïve mice. **D** Representative immunofluorescence colocalization of lineage reporters (green or red) with Krt5 (white) 14 days after PR8 infection, demonstrating Lin$^{3a2-high}$ and Lin$^{3a2-low}$ epithelial cells (1 experiment, *n* = 3 biological replicates). **E** Representative immunofluorescence colocalization of lineage reporters (green or red) with Krt5 (white) 14 days after PR8 infection, demonstrating recruitment of Lin$^{3a2}$-low epithelial cells into damaged alveolar epithelium (1 experiment, *n* = 3 biological replicates). **F** Representative immunofluorescence colocalization of tdT (Lin$^{3a2}$; red) with Krt5 (green) among lineage-positive alveolar clusters 21 days after PR8 infection (1 experiment, *n* = 3 biological replicates). **G** Experimental design for assessing lineage tagged populations using extended washout period. DR mice (*n* = 5 per experimental group) were treated with three doses of tamoxifen, infected with PR8, and recovered for the indicated time points. **H** Representative immunofluorescence colocalization of lineage reporters (red) with Krt5 (green) 14 and 21 days after PR8 infection using extended washout periods for tamoxifen (1 experiment, *n* = 5 biological replicates for all

except 14 dpi DR, for which *n* = 6). **I** Experimental design for the generation of in-vitro organoids from tdT-lineage tagged cells (*n* = 5). **J** Representative scatter plot of tdT and eGFP lineage-labeled cells from tissue homogenate of DR mice. Lineage reporter (tdT; eGFP) was assessed as a function of surface expression of CD326$^+$Sca1$^+$ airway epithelium and CD326$^+$Sca1$^-$ alveolar epithelium. **K** Representative immunofluorescence colocalization of lineage reporters (Lin$^{3a2}$; red) with Krt5 (green) from organoid cultures demonstrating Lin$^{3a2-high}$ and Lin$^{3a2-low}$ epithelial cells. **L** Quantification of recombination efficiency following immunofluorescent staining for tdT, eGFP, and Scgb3a2 (as in 'D'; tamoxifen washout period = 9 days). Red bars show either Scgb3a2 or lineage traced (either tdT or eGFP) cells per unit basement membrane. Black bar represents the fractional representation of Scgb3a2-immunofluorescent cells that are lineage-positive. Data are presented as mean values ± SEM. *n* = 3 biologically independent samples per condition, with significance determined by Mann–Whitney two-tailed *U*-test. Source data are provided as a Source Data file. **M** Contribution of lineage-tagged populations to hBC in alveolar 'pods' as a function of time (as in 'H'; tamoxifen washout period = 28 days) after PR8 infection. Data are presented as mean values ± SEM. *n* = 5 per condition, with significance determined by Mann–Whitney two-tailed *U*-test (**P* < 0.01). Source data are provided as a Source Data file. **N** Quantification of flow cytometry (as in 'J'; tamoxifen washout period = 28 days) from tissue homogenate from PR8-infected *DR* mice. Lineage reporter (tdT; Lin$^+$) was assessed as a function of surface expression of Sca1 and data was presented as the fraction of positive cells. Data are presented as mean values ± SEM. *n* = 4 biologically independent samples per condition, with significance determined by Mann–Whitney two-tailed *U*-test (**P* < 0.05). Source data are provided as a Source Data file.

infection, no differences were observed in the fate of either IS or club cells with either 9 day or 28-day tamoxifen washout periods. Consequently, 9 or 28-day tamoxifen washout periods were used interchangeably for PR8 experiments. Tamoxifen exposure was followed by a wash-out period of 9 days before analysis of lineage tracing in lungs of naïve mice (Fig. 3B). Control experiments involving TAM exposure of mice harboring *Scgb3a2-DreER* and Dre/Cre reporter but lacking *Scgb1a1-CreER*, revealed efficient induction of tdT within Scgb3a2-expressing airway cells without evidence of eGFP reporter expression, confirming the specificity of DreER for RoxP without recombination of LoxP sequences (Supplementary Fig. 3A–C). We went on to assess the steady-state phenotype of both IS and club cells in dual recombinase mice harboring *Scgb3a2-DreER*, *Scgb1a1-CreER*, and Dre/Cre recombinase reporter alleles. Lineage labeling of Scgb3a2 immunoreactive cells in the airways of naïve mice was determined to be 75.2 ± 3.06%, of which Lin$^{3a2}$ IS cells (16.8% of total lineage-labeled cells) were interspersed among Lin$^{3a2/1a1}$ club cells (83.2% of total lineage-labeled cells) (Fig. 3C, D, L). Lineage-labeled (tdT$^+$) serous cells in the airways of naïve mice were the only cell type that showed positive immunofluorescent staining for Bpifa1 and were uniformly negative for markers of ciliated and BC (Fig. 3C). Notably, we found no evidence that IS cells include a rare p63-expressing subset (Supplementary Fig. 3D–G), suggesting that they represent a distinct pool of hBC progenitors from those previously proposed through lineage tracing using a Tp63-CreER driver allele[15,16]. We attribute this discrepancy to the consequences of Tp63 haploinsufficiency of the Tp63-CreER allele.

We next sought to assess the fate of lineage-labeled IS and club cells following PR8-induced lung injury. At the 14-day post-PR8 recovery time point, both Lin$^{3a2/1a1}$ and Lin$^{3a2}$ cells were observed in the airways (Fig. 3D, E). Whereas Lin$^{3a2/1a1}$ cells occupied a luminal location and were exclusively Krt5$^-$, Lin$^{3a2}$ cells occupied both luminal and basal locations with basal-localized cells showing positive Krt5 immunoreactivity. The observed behavior of Lin$^{3a2/1a1}$ cells is consistent with prior reports by others and us that club cells do not contribute significantly to the expanding pool of BC in response to lung injury or infection[9,14]. Interestingly, Lin$^{3a2}$ cells included both tdT-bright (Lin$^{3a2-high}$) and tdT-dim (Lin$^{3a2-low}$) populations, this property correlating with Krt5$^-$ and Krt5$^+$ immunoreactivity, respectively. Injured

alveolar regions were repopulated exclusively by basal-like Krt5-immunoreactive cells that were Lin$^{3a2-low}$ at day 14 post-infection (Fig. 3E). By recovery day 21, Lin$^{3a2}$ cells occupying alveolar regions included Krt5$^+$Lin$^{3a2-low}$, Krt5$^+$Lin$^{3a2-high}$, and Krt5$^-$ Lin$^{3a2-high}$ patches, with some patches appearing to show mixed phenotypes suggestive of transitioning cell states (Supplementary Fig. 3H, I; Fig. 3F).

Control experiments with an extended washout period (28 days) were performed to ensure no residual tamoxifen-mediated recombination that could confound the interpretation of lineage tracing experiments. Results with a 28-day washout mirrored those using a 9-day washout, suggesting that washout periods of 9 days or greater, under the conditions of tamoxifen preparation and delivery used in this study, were sufficient to faithfully assess IS/club cell fate after PR8 infection (Fig. 3G, H, M). Immunofluorescence staining for Scgb3a2 confirmed that cells phenotypically similar to IS cells were included among the alveolar Krt5$^-$Lin$^{3a2-high}$ population (Supplementary Fig. 3J). Furthermore, hBC did not show clear evidence of a differentiation trajectory towards alveolar epithelium (Supplementary Fig. 3K, L). These data are consistent with trajectory analysis of scRNAseq data suggesting that hBC observed after PR8 infection eventually yields differentiating airway epithelial cells such as Scgb3a2-expressing IS cells (Fig. 2C). Lin$^{3a2-high}$ cells displayed immunophenotypic characteristics of IS cells (Fig. 3C) and show increasing abundance with time after PR8 infection, consistent with predictions made by scRNAseq (Fig. 1E).

To further verify that both Lin$^{3a2-high}$ and Lin$^{3a2-low}$ epithelial cells were derived from Lin$^{3a2-high}$ IS cells, Lin$^{3a2}$ IS cells were FACS enriched from dissociated lung tissue and evaluated their clonal potential in 3D organoid assays (Fig. 3I, J, N). Consistent with the observed behavior of Lin$^{3a2-high}$ IS cells in vivo, IS-derived organoids recapitulated the pseudostratified epithelial structure of conducting airways including both Lin$^{3a2-high}$/Krt5$^-$ and Lin$^{3a2-low}$/Krt5$^+$ epithelial cells (Fig. 3K). Our data confirm that both Lin$^{3a2-high}$ and Lin$^{3a2-low}$ cells are derived from a common Lin$^{3a2-high}$ IS progenitor and that changes in reporter fluorescence intensity correlate with alterations in cell state. Taken together, the changing milieu of the PR8 injured lung initially results in the specification of hBC from an airway IS progenitor, followed by their differentiation back to IS and other luminal cell types in airways and injured alveolar regions.

## Regulatory gene networks involved in immune-epithelial crosstalk are activated in nascent basal but not in IS cells

Next, we sought to define mechanisms that regulate the specification and fate of hBC observed in the lungs of PR8-infected mice. To gain further insights into pathways regulating the behavior of either IS or BC, we evaluated changes in regulatory genes over the time course of PR8 infection and recovery using Bigscale2[32,33]. ScRNAseq data from naïve and days 11–17 after PR8 infection were sorted by cell type to yield four regulatory gene networks (GRN; Supplementary Fig. 4A). Cell type-specific GRNs from post-infection time points were then compared to their respective naïve cell controls to normalize node and edge values (Supplementary Fig. 4B). A gene list of the top 300 delta degree node centralities was generated and used to identify enriched gene ontology (GO) terms by Panther-based overrepresentation tests (Supplementary Fig. 4C; Fig. 4A). Gene regulatory networks shown in Supplementary Fig. 4C demonstrates dynamic changes occurring among BC but less so for IS cells between naïve and post-infection time points. Highly enriched GO terms in basal populations included pathways involving cell proliferation, such as activation of canonical Wnt signaling[34] consistent with observed BC proliferation (Fig. 2A) and associated hyperplasia (Fig. 1D) that accompanies recovery from PR8 infection. Pathways associated with cytokine signaling and innate immune stimulation of epithelial cells were also significantly upregulated in BC, and to a lesser extent in IS cells, isolated from PR8-infected lungs (Fig. 4A).

Innate immune activation is a well-documented response to respiratory viral infection in general and is a key regulator of epithelial cell fate leading to remodeling in airways. Notably, local production of TNF and IL-1β following influenza virus infection in mice promotes the regenerative capacity of alveolar epithelium[18], and IL-22 enhances survival and reduces lung fibrosis following PR8 infection[19]. To explore further the potential roles for cytokine-mediated changes in epithelial cell fate, we used a multiplex immunoassay to quantify cytokine responses in the lungs of influenza virus-infected C57/Bl6 mice. We observed a significant increase in interferon (Ifnγ) and pro-inflammatory cytokines IL-6, TNF, and IL-1β (Fig. 4B; Supplementary Fig. 4D), consistent with known responses to respiratory viral infection[18]. Significant induction of type 17-related cytokines, including IL-17a and IL-22, was observed in lung tissue homogenates and/or bronchoalveolar lavage (BAL) of mice between days 5–11 after PR8 infection (Fig. 4B, Supplementary Fig. 4D). The protective effects of IL-22[19] and the coincidence of increased cytokine production with BC expansion in airways and alveoli led us to speculate that IL-22 signaling plays a role in BC expansion following PR8 infection.

## Renewal and differentiation of hBC is regulated by γδT-cell-derived IL-22

To further explore roles for innate immune activation and IL-22 in the regulation of epithelial cells in the airways and alveoli of PR8-exposed mice, we used scRNAseq to define changes to immune cell populations elicited in response to infection (Fig. 4C; Supplementary Fig. 5A). ScRNAseq data were collected from CD45+ lung cells recovered at different times following PR8 infection. UMAP dimensional reduction of aggregated data allowed visualization of major immune subsets (Fig. 4D, E). We sought to define the spatial context of these major immune subsets in relation to hBC observed during recovery. Gene signatures derived from immune enriched scRNAseq dataset (Fig. 4D; Supplementary Fig. 5A) were used to infer cell localization within spatial RNAseq data (Fig. 4F). Spatial gene expression analysis indicates preferential colocalization of γδT cells within BC-rich regions in recovering lung tissue (Fig. 4F). Interestingly, type 17 γδT cells show increased abundance within regions of BC hyperplasia (Fig.4F, G). This led us to consider the possible regulatory influence of γδT lymphoid subset over BC fate.

Since several immune subsets can secrete IL-22, we generated mice harboring *IL-22^Cre*/ROSA-26-tdT* to fate map IL-22-lineage immune cells during the repair response to PR8 infection (Fig. 5A). To identify the cell types expressing IL-22, we gated IL-22 lineage tag (tdT) CD45+ cells for expression of lymphoid markers CD3 and CD4 (Fig. 5B). We found that CD3+CD4− T cells subsets represented the bulk of IL-22-lineage immune cells elicited in response to PR8 infection, with minor contributions made by CD3+CD4+ Th17 cells (Fig. 5B, Supplementary Fig. 6A). Non-T lymphoid CD3- subsets did not contribute to the IL-22-expressing lineage following PR8 infection (Supplementary Fig. 6A). Immunofluorescence analysis demonstrated that IL-22 and γδTcr-immunoreactive immune cells localize to hBC and were induced in response to acute lung injury (Fig. 5C–E).

To identify epithelial cell types capable of responding to IL-22 signaling, we use immunofluorescence to identify which epithelial subsets express IL-22ra1, the subunit of the heterodimeric IL-22 receptor that confers IL-22 ligand specificity. Notably, within the conducting airway of naïve mice, the highest level of IL-22ra1 immunoreactivity was observed among post-mitotic ciliated epithelial cells, with little to no immunoreactivity seen among club, basal, or serous cells (Fig. 5F). However, a significant level of IL-22ra1 expression was observed in alveolar hBC (hereon called pods[13]) during recovery following PR8 infection (Supplementary Fig. 7 A; Fig. 5G). Interestingly, the intensity of IL-22ra1 staining was greatest among hBC located at the periphery of pod regions, suggesting that they harbor the capacity to respond to IL-22 elicited in response to PR8-induced injury (Supplementary Fig. 7B). Taken together, our data suggest a process of altered IL-22 responsiveness exclusively among nascent basal populations of the PR8-injured lung that regulate BC fate.

## IL-22 promotes self-renewal of hBC, allowing hyperplastic expansion in airways and injured alveoli

To investigate the contribution made by IL-22 in the regulation of BC fate after PR8 infection, we used *IL-22^Cre/Cre* (IL-22 LOF) and *IL-22ra1^fl/fl/Shh-Cre* (IL-22r cLOF) mice to modulate IL-22 levels and signaling, respectively (Fig. 6A). Compared to their corresponding WT control groups, neither IL-22 LOF nor IL-22r cLOF mice showed significant differences in weight loss, viral gene expression, or loss of parenchymal Pdpn immunofluorescence, following PR8 infection (Supplementary Fig. 8A–C; Fig. 6B). Next, we assessed expansion of Krt5-immunoreactive BC as a function of damaged area in IL-22 LOF mice following PR8 infection. A reduction in Krt5+ BC hyperplasia was observed in IL-22 LOF mice compared to WT control mice at all time points examined, which reached statistical significance by the day 17 post-infection time point (Fig. 6D). These data were confirmed by assessing *Krt5* mRNA content within total lung RNA isolated at each timepoint, for which statistically significant declines in *Krt5* mRNA were observed at both 14 day and 17 day post-infection time points (Fig. 6E). Similar observations of reduced BC expansion were seen among IL-22r cLOF mice following PR8 infection (Supplementary Fig. 8C).

Similarly, in other tissues and carcinomas, IL-22 regulates the proliferative potential of epithelial stem progenitors and tumor cells[35]. To assess if reduced BC hyperplasia observed in IL-22 LOF mice was associated with reduced epithelial proliferation, we measured the BC proliferative index by immunofluorescence of Ki67 and Krt5 following PR8 infection. The Ki67-labeling index of BC observed at 7 days post PR8 infection did not differ between IL-22 LOF, IL-22r cLOF, and their corresponding WT controls (Fig. 6C, F). These data indicate that BC expansion at early time points after PR8 infection occurs in an IL-22-independent manner and is consistent with the observed lack of IL-22ra1-immunoreactivity in airway BC of either naïve control mice or mice at early time points after PR8 infection (Fig. 5F). However, hBC observed within airway and alveolar regions of both IL-22 LOF and

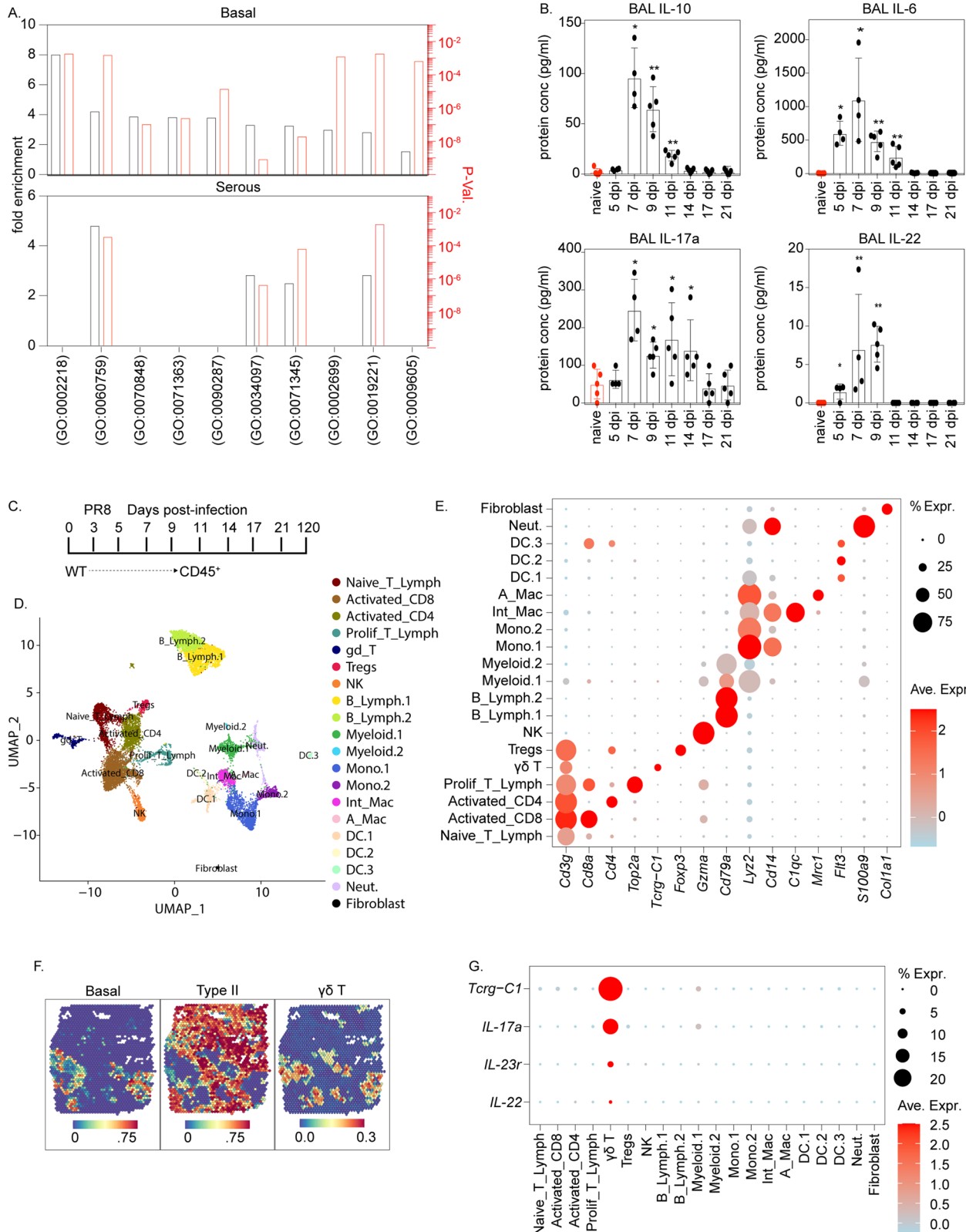

IL-22r cLOF mice at 14 days post-PR8 infection showed significantly reduced levels of Ki67 staining compared to their corresponding WT controls (Fig. 6C, F). These data indicate that at later post-exposure time points, IL-22 promotes BC expansion and self-renewal, resulting in the maintenance of proliferative potential. All genotypes showed reduced BC Ki67 proliferative indices that did not differ significantly between groups at 21 days post-PR8 infection (Fig. 6C, F).

To determine if the lack of IL-22 stimulation impacts the fate of alveolar BC following PR8 infection, we co-stained pod regions observed in parenchymal tissue of PR8-infected WT and IL-22 LOF mice for BC and IS markers. Interestingly, Krt5-immunoreactive pods observed in lungs of either IL-22 LOF or IL-22r cLOF mice 14 days post-PR8 infection displayed evidence for Scgb3a2 expression that was absent in comparable Krt5-immunoreactive pod structures of wildtype

**Fig. 4 | T lymphoid cells colocalize with hBC at sites of lung injury following PR8 infection. A** Representation of significant GO terms associated with immune > epithelial interaction enriched among PR8-infected BC (upper) and IS cells (lower) using a Panther overrepresentation test. Source data are provided as a Source Data file. **B** Assessment of selected cytokine levels in BALF during PR8 infection. Data are presented as mean values ± SEM. $n = 4$ for the 7-day post-PR8 condition. $n = 5$ For all other conditions. All analyzed samples were biologically independent. Statistical significance was determined by Mann–Whitney two-tailed $U$-test (*$P < 0.05$, **$P < 0.01$). Source data are provided as a Source Data file. **C** Experimental design. Mouse lung homogenates were collected from 8 to 12-week-old C57/Bl6 mice. CD31⁻CD326⁺CD45⁺ immune cells were enriched by FACS and transcriptomes were assessed by scRNAseq ($n = 5$ per group). **D** UMAP plot of combined scRNAseq data. Unsupervised clustering was used to distinguish distinct cell phenotypes which were assigned to known immune cell types based upon gene signatures. **E** Dot Plot selected cell-type-specific gene expression for each annotated cluster. **F** Cell type prediction scores of immune populations represented within spatial RNAseq data. Cell type-specific gene signatures were generated using immune scRNAseq data in Fig. 5A and epithelial scRNAseq data in Fig. 1A. Color scale at the bottom reflects the intensity of cell type prediction scores. **G** Dot plot showing activation of type-17 gene signature (IL-17a, IL-23r, and IL-22) in γδT cells.

control mice (Fig. 6G). The appearance of pod structures composed of epithelial cells showing colocalization of both Krt5 and Scgb3a2 immunofluorescence is consistent with earlier data showing BC > IS re-differentiation, and supports the notion that residual Krt5-immunoreactivity, in light of the long half-life of intermediate filament proteins[36], reflects the BC origin of differentiating IS cells. To further examine the impact that loss of IL-22 has on BC maturation, scRNAseq profiles were generated for Sca1⁺ airway enriched cells isolated from lungs 21 days after PR8-infection of either IL-22r cLOF mice or their corresponding WT controls (Fig. 6H). Reduced BC numbers in the lungs of IL-22r1a-cLOF mice compared to WT control mice were confirmed by UMAP clustering (Fig. 6I). A significant decrease in the abundance of epithelial cells expressing BC marker genes, was observed among IL-22r1a-cLOF mice compared to their WT controls (Fig. 6J). Collectively, our data suggest that IL-22 functions to promote BC renewal over differentiation and that downregulation of innate immune responses at late time points following PR8 infection with associated reduction in IL-22, serves as a trigger to promote basal to serous cell differentiation.

## Discussion

Basal cell (BC) hyperplasia and colonization of the injured alveolar gas-exchange region are significant determinants of tissue remodeling and morbidity among patients with severe respiratory viral infections and share features of distal lung remodeling in patients with interstitial lung disease. Here we show that hBC elicited by infection of the mouse respiratory tract with H1N1 influenza virus (strain PR8) is derived predominantly from a serous cell subset of intralobar secretory cells. Hyperplastic BC were distinguished from pre-existing BC by their relatively immature molecular phenotype and expression of IL-22Ra1. Innate immune activation and associated secretion of IL-22 promoted self-renewal of hBC in airways and establishment of hyperplastic foci within injured alveoli. Germline loss of IL-22 or conditional loss of IL-22ra1 expression within epithelial cells, limited expansion and promoted premature differentiation of hBC into serous cells. These findings establish a mechanism that promotes the expansion of serous cells and their colonization of injured alveolar epithelium, leading to epithelial remodeling and loss of normal alveolar epithelium following respiratory viral infection.

We provide evidence for the existence of two independent secretory cell lineages within intralobular airways of the mouse lung, club cells, and intralobular serous (IS) cells. Previous lineage tracing studies have demonstrated that Scgb1a1-expressing club cells are capable of unlimited self-renewal and replacement of specialized epithelial cell types of bronchiolar airways in mice[37]. However, epithelial cell injury in the lungs of PR8-infected mice leads to the activation of non-canonical epithelial progenitors leading to BC hyperplasia[8,9,14–16]. We show that only the IS subset of secretory cells can assume BC fates in the setting of severe respiratory viral infection, suggesting that IS cells function as a reserve epithelial progenitor that is unlikely to make significant if any contribution to homeostatic epithelial maintenance in distal airways. Expansion of IS cells following influenza virus infection and their phenotypic conversion to hBC suggest an unappreciated role

for IS cells in epithelial maintenance and repair following severe injury. Interestingly, IS cells labeled using the dual recombinase lineage-tracing approach outlined in our study showed no evidence of p63 immunoreactivity, suggesting that their contribution to basal cell expansion following PR8 infection is distinct from those of rare p63-lineage epithelial cells described in lineage tracing studies using a *Trp63-CreER* driver allele[15,16]. IS cells were among a heterogeneous population of H2-k1 high epithelial cells observed in lungs of either naïve or PR8-infected mouse lungs, raising the possibility that they represent a more defined subset of H2-k1 cells proposed as BC progenitors[38].

Our findings also shed further light on the biological significance of earlier work by Tata et al. who demonstrated that SSEA⁺ secretory cells of tracheobronchial airways have the unexpected capacity to replenish basal stem cells in a genetic model of BC ablation[2]. However, even though IS cells identified in our study function in a similar capacity to SSEA⁺ secretory cells defined by Tata et al. to yield hBC, we show that the fate/stemness of IS-derived BC is dictated by the regional microenvironment. Expansion of IS-derived hBC and subsequent differentiation led to proximalization of distal conducting airways through replacement of the normally simple cuboidal bronchiolar epithelium with a pseudostratified epithelium, thus demonstrating their multipotency in the airway microenvironment like that described for SSEA⁺ secretory cell-derived BC in the Tata et al. study. Furthermore, serous cells significantly expanded in number immediately during the infectious/inflammatory early phase following PR8 infection and showed significant up-regulation of interferon-responsive genes suggesting roles for interferon signaling in the regulation of serous cell expansion and phenotype. Serous cells were the predominant source of hBC colonizing injured alveolar regions, with hBC undergoing further proliferative expansion under the influence of IL-22. Hyperplastic BC colonized injured parenchymal tissue, followed by re-differentiation into IS cells for replacement of normal alveolar epithelium with a dysplastic serous cell-predominant epithelial lining. It is likely that the injured milieu and associated activation of innate immune responses following PR8 infection played a significant role in the magnitude of BC hyperplasia in our study compared to that resulting from targeted ablation of resident BC in tracheobronchial airways in the study by Tata and colleagues. However, PR8 infection of mice may more closely reflect lung injury, inflammation, and defective repair, seen in patients with either respiratory viral infection or chronic lung injury seen in end-stage IPF. Our data may therefore provide insights into mechanisms of epithelial remodeling observed in distal lung tissue of patients with idiopathic pulmonary fibrosis, where alveolar epithelial progenitor cell dysfunction is associated with BC hyperplasia in small airways and establishment of dysplastic cysts in place of normal alveolar epithelium[7,10,39].

Even though hBC elicited by PR8 influenza virus infection share many properties of pre-existing BC of pseudostratified airways, their immature molecular phenotype and unique expression of IL-22ra1 impart distinctive functional properties that promote rapid epithelial replacement in airways and alveoli. We found that IL-22-expressing γδT-cells are recruited to sites of PR8-induced airway and alveolar

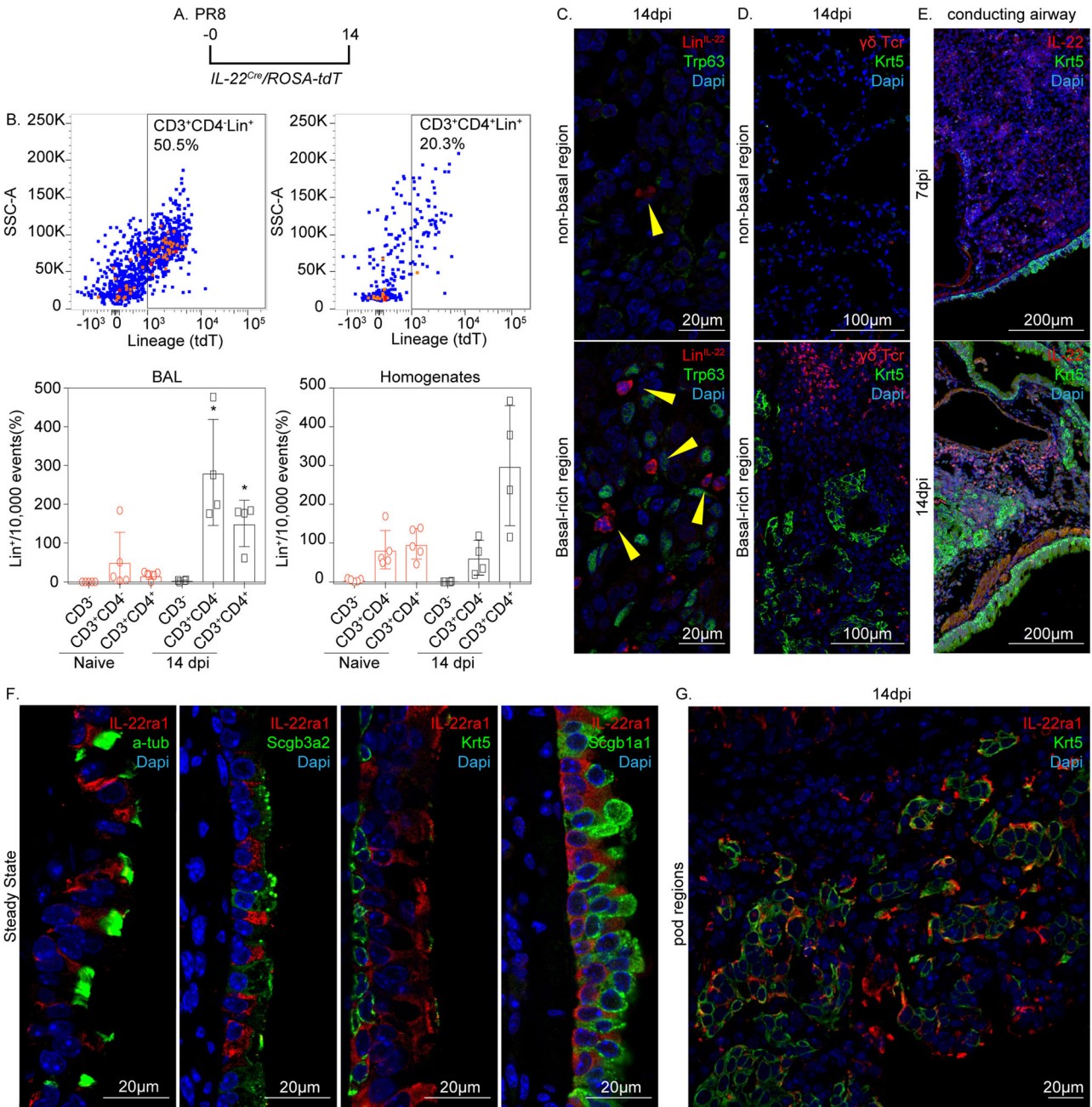

**Fig. 5 | Identity of IL-22 expressing cells and their localization to regions of BC hyperplasia in lungs of PR8 infected mice. A** Experimental design for fate mapping of IL-22 expressing cells following PR8 infection using *IL-22^Cre^/ROSA-tdT* mice. (**B**) Flow cytometry analysis of either BAL or tissue homogenate from PR8-infected *IL-22^Cre^/ROSA-tdT* mice. IL-22 lineage reporter (tdT; Lin⁺) was assessed as a function of surface expression of CD3 and CD4 and data was presented as the fraction of positive cells. Data are presented as mean values ±± SEM. *n* = 5 per naïve condition and *n* = 4 per PR8 condition. All analyzed samples were biologically independent. Significance was determined by Mann–Whitney two-tailed *U*-test (*$P < 0.05$). Source data are provided as a Source Data file. **C** Immunofluorescence detection of Lin⁺ cells as a function of p63-immunoreactive BC within BC-rich vs. BC devoid alveolar epithelium of PR8-infected mice (1 experiment, *n* = 5 biological replicates). **D** Immunofluorescence detection of γδTcr⁺ cells as a function of Krt5-immunoreactive BC within BC-rich vs. BC devoid alveolar epithelium of PR8-infected mice (1 experiment, *n* = 5 biological replicates). **E** Immunofluorescence localization of IL-22 expressing cells 7 and 14 days post-PR8 infection (1 experiment, *n* = 5 biological replicates). **F** Immunofluorescence colocalization of IL-22ra1 (red) with either a-tub (ciliated), Scgb3a2 (serous/club), Krt5 (BC), Scgb1a1 (club) (green) in airways of naïve mice (1 experiment, *n* = 5 biological replicates). **G** Representative immunofluorescent colocalization of IL-22ra1 (red) and Krt5 (green) in BC-rich alveolar region of PR8-infected mouse lung (1 experiment, *n* = 5 biological replicates).

injury and that their local production of IL-22 promoted self-renewal of hBC with no impact on either their specification in airways or migration to injured alveoli. Notably, either germline loss of IL-22 or conditional loss of IL-22ra1 within lung endoderm resulted in premature differentiation of alveolar BC into Scgb3a2⁺ serous cells, thus limiting BC hyperplasia. These data shed light on mechanisms of fibrosis in the

lungs of IL-22⁻/⁻ mice following PR8 infection[19], where IL-22 restrains hBC in a highly proliferative and migratory state allowing expansion and re-epithelialization of injured airways and alveoli.

We propose a model in which IS cells, through a BC intermediate, ultimately colonize injured alveoli that have been denuded of an epithelial lining following PR8 infection. Through a combination of

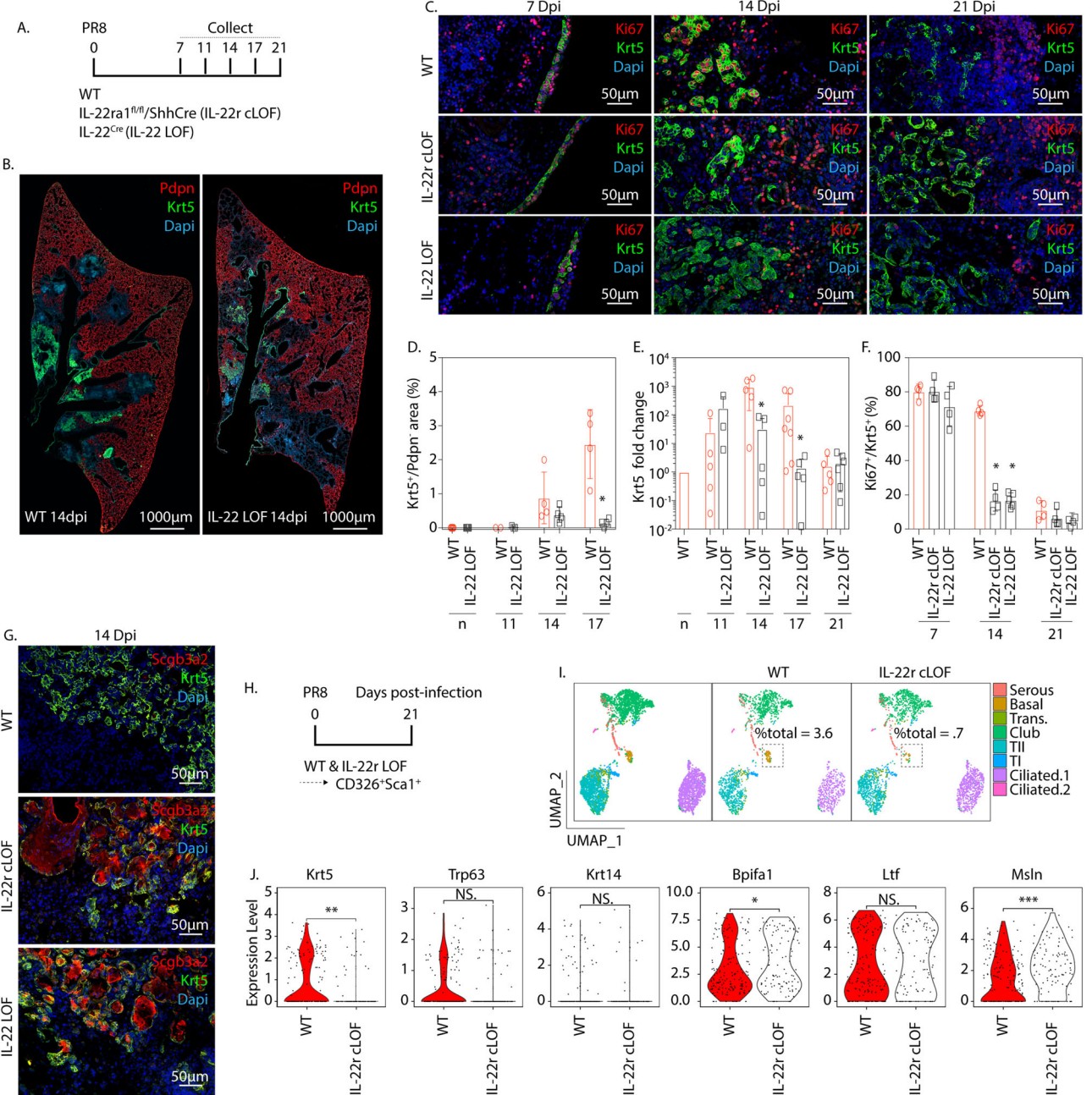

**Fig. 6 | IL-22 regulates BC fate in the PR8-injured lung. A** Experimental design to assess BC expansion in *IL-22^Cre* homozygous (IL-22 LOF) mice. Left lobes were collected for immunostaining and total RNA was isolated from the right cranial lobe for gene expression analysis. **B** Representative immunofluorescence localization of Pdpn (red) and Krt5 (green) in lungs of WT and IL-22 LOF mice 14 days post-PR8 infection. **C** Representative immunofluorescence localization of Ki67 (red) and Krt5 (green) in lung tissue of PR8-infected WT, IL-22 LOF, and *IL-22ra1^fl/fl/Shh-Cre* (IL-22r cLOF) mice. **D** Quantification of (B). The fraction of the Krt5 stained area was normalized to the area of the damaged alveolar epithelium (DAPI⁺Pdpn⁻). Data are presented as mean values ± SEM. *n* = 5, 3, 4, 4 for naïve, 11dpi, 14dpi, and 17dpi conditions, respectively). All analyzed samples were biologically independent. Statistical analysis was performed by Mann−Whitney two-tailed *U*-test (*P* < 0.05). Source data are provided as a Source Data file. **E** qPCR detection of *Krt5* mRNA in total lung RNA of PR8-infected WT and IL-22 LOF mice. Data are presented as mean values ± SEM. *n* = 5, 3, 5, 5, 7, 5, 5, 6 for 11dpi WT/KO, 14dpi WT/KO, 17dpi WT/KO, 21dpi WT/KO conditions, respectively. All analyzed samples were biologically independent. Statistical analysis was performed by Mann−Whitney two-tailed

*U*-test (*P* < 0.05). Source data are provided as a Source Data file. **F** Quantification of (**C**). Shown is the Ki67-labeling index for BC, expressed as a percentage of Ki67⁺, Krt5⁺ cells within areas of BC hyperplasia.). Data are presented as mean values ± SEM. *n* = 5 per IL-22 KO 14 dpi condition. *n* = 4 for all other conditions. All analyzed samples were biologically independent. Statistical analysis was performed by Mann-Whitney two-tailed U-test (*P* < 0.05). Source data are provided as a Source Data file. **G** Representative immunofluorescence localization of Krt5 (green) and Scgb3a2 (red) in lungs of PR8-infected WT, IL-22 LOF and IL-22r cLOF mice.

**H** Experimental design for scRNAseq of PR8-infected IL-22r cLOF mice (*n* = 5 biological replicates per group). **I** UMAP plot of Sca1⁺ airway enriched cells from IL-22r cLOF and C57/Bl6 control. Cell type annotation of clusters was performed based upon the expression of cell type-specific genes. UMAP plot were re-clustered based on genetic permutation to better represent changes in BC populations between experimental conditions. **J** Assessment of selected basal (*Krt5, Krt14, Trp63*) and serous (*Bpifa1, Ltf, Msln*) genes between IL-22r cLOF and WT control. Statistical analysis was performed by Wilcox Test (*P* < 0.05, **P* < 0.01, ***P* < 0.001).

IS > BC > IS phenotypic plasticity, we posit that the suppressive effect of IL-22 signaling on lung fibroproliferative responses to PR8 infection results from its influence on BC expansion and arrested differentiation. Interestingly, these roles for IL-22 in regulating PR8-elicited BC in the lung contrast with observations made in other organ systems, such as in the epidermis, where IL-22 promotes fibroproliferative responses associated with wound closure[40], and in the intestine, where IL-22 regulates epithelial cell fate and host defense through induction of antimicrobial factors[41,42]. We attribute these differences in outcome to target cells that respond to local production of IL-22 and to the impact of non-cell-autonomous influences of IL-22 signaling within epithelial and stromal cell types. Interestingly, in other prior studies, IL-22 has been shown to protect against influenza virus-induced pneumonia[43] and has been implicated in the production of antimicrobial factors that protect against secondary bacterial colonization following influenza virus infection[44]. Although early hBC migration and colonization of injured alveoli is likely beneficial to overall survival due to restoration of epithelial barrier function, proximalized parenchymal tissue observed in patients recovering from severe respiratory viral infections[11,12] and in lung tissue of patients with end-stage IPF[7,10,45–47], suggest a potentially pathologic effect through reduction of normal alveolar surface area and an associated decline in lung diffusing capacity. Our data demonstrate that these effects are an indirect consequence of IL-22-mediated expansion of IS-derived BC followed by their re-differentiation to yield IS hyperplasia and provide insights into mechanisms of protection against secondary bacterial pneumonia after respiratory viral infection. These findings suggest that pharmacologic targeting of IL-22 signaling may suppress BC expansion and parenchymal remodeling in lung diseases associated with proximalization of the injured alveolar epithelium, such as following severe respiratory viral infection or in patients with chronic interstitial lung disease.

## Methods

### Ethics statement
The studies involving mice were performed in strict accordance with the recommendations in the Guide for the Care and Use of Laboratory Animals of the National Institutes of Health. All studies using mice were performed using protocols that were approved by the Cedars-Sinai Institutional Animal Care and Use Committee and by the Institutional Biosafety Committee.

### Mouse strains
*Sftpc-CreER*[48], *Scgb1a1-CreER*[37], *Krt5-CreER*[49], *IL-22*[Cre] mice[50], and *Shh-Cre*[51] were obtained from Jackson Labs. *IL-22ra1*[fl/fl 52] were provided by Jay Kolls. These mice were bred to either *ROSA-26-mTmG*[53], *ROSA-26-tdTomato* (supplied by D. Jiang and P.W. Noble) or *RC::RLTG*[54] mice for lineage-tracing experiments. Mice were maintained as a specific pathogen-free colony in barrier housing, with controls being either co-housed siblings or independently bred lines depending upon genotype and experimental parameters. All experiments were performed using 8–12-week-old mice of mixed gender and generated using a C57BL/6J background. Mice were euthanized via intraperitoneal injection of ketamine/xylazine followed by cervical dislocation. An equal distribution of male and female mice was selected for each experiment.

*Scgb3a2-DreER* mice were generated by Jackson Labs using a CRISPR/Cas9 knock-in strategy. Mice were produced by inserting an IRES-DreERT2 construct into the 3' UTR of the mouse *Scgb3a2* gene resulting in tamoxifen-inducible DRE recombinase activity under the control of the *Scgb3a2* promoter. Four Founders were identified following embryotic manipulation that carried the desired genetic permutation which was confirmed through long-range qPCR. Founders were backcrossed to Bl6/J mice to generate N1 progeny which were then bred to mice containing at least one copy of both *Scgb1a1-CreER*

and *RC::RLTG* to generate the first generation of experimental mice for in vivo fate mapping experiments.

### Tamoxifen
Tamoxifen was dissolved by sonication at a concentration of 20 mg/ml in corn oil until clear and stored at −80 °C in 50 ml aliquots. A washout of 9 days following the last treatment with TM was used for initial lineage tracing analysis prior to infection. For these experiments, DR mice were injected intraperitoneally using a 27 gauge needle three times at a dose of 0.225 mg/g mouse body weight. Additional experiments were performed on DR mice with extended tamoxifen washout, to confirm that lineage tracing experiments were not confounded by residual tamoxifen and associated nuclear Cre recombinase activity following exposure to the virus. For these experiments, the tamoxifen washout period was extended to 28 days and mice were treated by oral gavage at a dose of 0.225 mg/g body weight. No differences in lineage tracing outcome were observed between animals undergoing 9-day vs. 28-day tamoxifen washout prior to PR8 infection.

### Influenza virus inoculation
A mouse-adapted variant of the 1918 H1N1 influenza (Puerto Rico 8; PR8) was used to infect mice and induce acute lung injury. A single treatment of 100PFU/50 µl was administered intratracheally into mice placed into a surgical plane through inhalation exposure to isoflurane. Mouse weight was monitored over a 14-day period to verify uniformity of infection between individuals. Expression of viral M1 and M2 gene transcripts was assessed from lung homogenates of representative mice to further confirm that a reproducible level of infection has occurred under all experimental conditions and between mouse genotypes. Mouse lung biopsies were collected at time points that correspond with hyperplasia of basal-like cells in the distal lung tissue of PR8-infected mice.

### Preparation of single-cell suspensions for single-cell RNAseq
**Type II and immune subsets.** Mouse lung biopsies were collected at the indicated time points: naïve, 3, 5, 7, 9, 11, 14, 17, 21, 60, 120, and 240 days post-infection. A sample size of 5 C57/Bl6 WT mice was used for each timepoint. Cell suspensions from each condition were pooled together prior to cell sorting. On the day of biopsy collection, the entire mouse lung was separated from the chest cavity and stored in a conical containing 4 °C 1XHBSS. Isolated lung lobes were intratracheally instilled with a 3 ml mixture containing 1 U elastase/1 ml 1XHBSS for 30 min at 37 °C. The crude cell suspension underwent mechanical agitation prior to incubation in dissociation solution with a final composition of 1X Liberase/1X HBSS for 30 min at 37 °C. The dissociation buffer was quenched with a solution containing 2% FBS/1 mM EDTA/1X HBSS on ice. Cells were filtered through a 70 µm nylon mesh to remove undigested tissue. Cells were centrifuged at 500×*g* for 10 min and resuspended in 1 ml blood cell lysis solution for 1 min to remove red blood cells from the suspension. Red blood cell lysis buffer was quenched using 25 ml 2% FBS/1 mM EDTA/1X HBSS, followed by centrifugation at 500×*g* for 10 min to pellet intact cells. When isolating epithelium, cells were resuspended in 1 ml 2% FBS/1 mM EDTA/1X HBSS, and magnetic bed separation was performed to deplete CD31 and CD45 subsets. Cell suspensions were enriched for epithelial and immune populations by FACS, with selection for CD326⁺CD45⁻CD31⁻ and CD326⁻CD45⁺CD31⁻ to yield enriched epithelial and immune cell fractions, respectively. A minimum of 50,000 epithelial cells and 100,000 immune cells were sorted for each timepoint.

**Conducting airway subsets.** Single-cell RNA seq data was generated from cell suspensions enriched for conducting airway epithelium. The aforementioned cell suspension prep was performed on tamoxifen inoculated *Sftpc-CreER/ROSA-mTmG* and a gating strategy was used to enrich for eGFP negative epithelium (i.e. CD326⁺CD45⁻CD31⁻eGFP⁻) by

FACS. For these experiments, a sample size of 3 *Sftpc-CreER/ROSA-mTmG* was pooled together prior to cell sorting. Cells were collected at the same indicated time points (naïve, 3, 5, 7, 9, 11, 14, 17, 21, 60, 120, and 240 days post-infection) and a minimum of 50,000 cells were collected per timepoint. In experiments where isolation of conducting airway epithelium was not possible using *Sftpc-CreER/ROSA-mTmG* (Fig. 6H) an alternative gating strategy was used to deplete Type II cells. A combination of Sca-1 and CD24 antibodies was used in lieu of the tdT lineage tag (i.e. CD326⁺CD45⁻CD31⁻Sca1⁺CD24⁺).

## Bioinformatics of scRNAseq

**Dataset generation.** Isolated single-cell suspension was resuspended in 0.04% BSA at a concentration of 600 cells/μl and were used to generate GEMS using a 10x genomics chromium controller. TotalSeq Hashtag Antibodies (Biolegend:15580#) were used to multiplex samples from isolated conducting airway epithelium (i.e. those cells isolated from tamoxifen-exposed *Sftpc-CreER/ROSA-mTmG* mice described in the previous section). ScRNAseq datasets described in Figs. 1A and 4C were generated using Chromium Single Cell 3′ v2 chemistry. ScRNAseq dataset described in Fig. 6H was generated using Chromium Single Cell 3′ v3 chemistry. Libraries were generated as per manufacturer instructions. Sequencing was performed on a Novaseq 6000 using a S4 150 and 28 + 90 bp paired-end setting. Alignment was performed using 'Cellranger count' function provided in 10x Genomics single-cell gene expression software.)

**Secondary analysis.** The R package 'Seurat'[55,56] was used to apply standard quality control metrics and unsupervised clustering (FindClusters(x, resolution = 0.5) in Seurat) leading to the generation of initial UMAP projections. Cell types were annotated based on gene expression for canonical markers and saved into object metadata (object@meta.data$anno in Seurat). Single-cell datasets were subsetted based on 'cell type' annotation and changes in gene expression were assessed at different time points after PR8 exposure using the hallmark gene sets from the molecular signature database[57,58]. Cell type annotation was performed using the following cell type-specific markers: Ltf (Serous cells[59,60]), Krt5, Trp63 (Basal cells[61,62]), Cldn10, Scgb1a1 (Club cells[63,64]), Ager (Alveolar Type I cells[65]), Sftpc, Chil1, Hc, Scid1 (Alveolar Type II cells[66]), Foxj1 (Ciliated cells[67]). The R package 'Velocyto' and 'scVelo'[31,68] was used to perform the single-cell trajectory analysis. Fastq files were aligned to a separate reference transcriptome containing both splice and unspliced gene variants. Before calculating RNA velocity, cell types of interest were subsetted from the compiled epithelial dataset and further subsetted based on timepoint. This was primarily done to reduce the computation required by the Velocyto package. Degree node centralities were calculated from gene regulatory networks generated by the package 'Bigscale2'[32]. The 'Bigscale2' package has the functionality to calculate the difference in node centralities between two conditions and this was used to generate ranked lists of the top 300 delta-degree node centralities between early and late infection. The resultant rank list was used to assess enriched GO terms using a PANTHER-based over-representation test produced by the gene ontology consortium[69–71]. Gene networks were visualized using Cytoscape_v3.8.1. The R package 'FGSEA'[72] was used to perform gene set enrichment analysis.

**Spatial transcriptomics.** Frozen 10 μm sections from 14 days post-PR8-infected mouse lungs were placed within the frames of the capture areas on the active surface of the Visium spatial slide. Tissue sections were fixed in methanol and stained with H&E. Bright-field images of stained sections in the fiducial frames were collected in 40× fields using Zeiss Axioscan Z1 microscopy. Stained tissue sections were permeabilized for 30 min and mRNA was released to bind oligonucleotides on the capture areas. Single-cell RNA-seq libraries were prepared as per manufacturer instructions and sequenced on a Novaseq 6000 using SP 28 + 90 bp paired-end reads. Count matrices were generated using the 'spaceranger count' function in Space Ranger 1.0.0. The resulting data were processed in Seurat. Mouse scRNAseq clusters described in Figs.1A and 4C were transferred to a spatial transcriptomic sample. The transfer function generates a probability score for each spot and its association with a given scRNAseq cluster. The spot is assigned to the cluster with the highest score and mapped back to the spatial transcriptomic sample image.

## Immunofluorescent microscopy

**Immunofluorescence staining.** To prepare mouse lungs for histology, we inflation fixed freshly dissected mouse lungs through the instillation of 1 ml 4% paraformaldehyde directly into a cannulated mouse trachea. After incubating the lungs for 24 h, the left lobe from each mouse was separated into a labeled cassette and stored in either 1X PBS for immediate tissue processing or in 70% EtOH for long-term storage. Tissues were dehydrated using the ASP300 tissue processor(Leica). After processing, left lobes were embedded in paraffin wax and sectioned at 7–9 μm thickness using an HM 325 rotary microtome(Leica). Sectioned slides were dried at room temperature until needed. After the selection of an appropriate panel, tissue sections were deparaffinized using the Shandon veristain Gemini ES (Thermofisher, cat: A7800013). Slides were transferred into a reservoir containing either citrate or tris-based antigen retrieval solution(vector labs, cat:3300/3301) and heated using a pressure cooker (Biovendor, cat: RR2100-EU). The slides were then blocked with 2% BSA for 30 min. Tissue sections were incubated in primary antibody overnight at 4 °C. Cells were then washed 5 times with 1X PBS and incubated in a solution containing both alexaflour conjugated secondary antibody and DAPI for 2 h. Slides were cover-slipped using Fluromount-G (EMS, cat:7984-25). Images were taken using either the Zeiss 780 confocal microscope or the Zeiss Axioobserver Z1 inverted microscope.

**3D organoid cultures.** MLg cells (ATCC, cat: CCL-206) were expanded under recommended culture conditions and co-cultured in transwells containing matrigel (Corning, cat: 354234) with primary mouse lung epithelial cells. Cells were seeded at a density of 3000:50,000 for lung epithelial cells and MLgs respectively. Cells were cultured in the presence of SB (Tocris, cat: 431542) for 7 days, at which point, incubations occurred in the absence of SB for an additional 7 days. At day 14 matrigel plugs were removed from the transwell, paraffin-embedded, and sectioned using a microtome.

Following are primary antibodies used: Chicken polyclonal anti-eGFP (1:1000, Abcam, Ab13970); Chicken Polyclonal anti-Keratin 5 (1:500, BioLegend, 905901); Mouse monoclonal anti-eGFP AF488 conjugated (1:500, Santa Cruz, Sc-9996); Rat monoclonal anti- IL-22ra1 (1:200, R&D systems, MAB42341); Goat polyclonal anti-tdTomato (1:500, Sicgen, Ab8181-200); Goat polyclonal anti-p63(1:500, Santa Cruz, Sc-8609); Goat Polyclonal UGRP1/SCGB3A2 (1:1000, R&D Systems, AF3465); Syrian hamster Monoclonal anti-Pdpn (1:1000, Life-Span Biosciences, LS-C143022-100); Rabbit Polyclonal anti-SCGB1A1 (1:500, Proteintech, 10490-1-AP); Rabbit Polyclonal anti-tdT (1:500, Rockland, 600-401-379); Rabbit Polyclonal anti-Msln (1:500, Thermo Fisher Scientific, PA5-79698); Rabbit Polyclonal anti-Ltf (1:200, Thermo Fisher Scientific, PA5-95513); Rabbit Polyclonal anti-Bpifa1 (1:200, Sigma-Aldrich, AV42475); Rabbit polyclonal anti-Keratin 5 (1:500, Cell Marque, EP1601Y); Rabbit polyclonal anti-Keratin 5 (1:500, Santa Cruz, Sc-66856); Rabbit polyclonal anti IL-22 (1:200, Abcam, ab18499); Rabbit polyclonal anti-Ki67 (1:1000, Ebioscience, 14-5698-82).

Following are secondary antibodies used: Goat anti-Chicken Alexa Fluor 488(1:500, Thermo Fisher Scientific, 6100-30); Goat anti-Hamster Alexa Fluor 488 (1:500, Thermo Fisher Scientific, A-21110); Donkey anti-Rabbit Alexa Fluor 488 (1:500, Thermo Fisher Scientific, A-21206); Donkey anti-Goat Alexa Fluor 555(1:500, Thermo Fisher Scientific, A-21432); Donkey anti-Rabbit Alexa Fluor 555 (1:500, Thermo

Fisher Scientific, A-31572); Goat anti-Chicken Alexa Fluor 568 (1:500, Thermo Fisher Scientific, A-11041); Donkey anti-Rat Alexa Fluor 594(1:500, Thermo Fisher Scientific, A-21209); Goat anti-Hamster Alexa Fluor 594 (1:500, Thermo Fisher Scientific, A-21113); Goat anti-Chicken Alexa Fluor 647 (1:500, Thermo Fisher Scientific, A-21449); Donkey anti-Rabbit Alexa Fluor 647 (1:500, Thermo Fisher Scientific, A-31573); Donkey anti-Goat Alexa Fluor 647 (1:500, Thermo Fisher Scientific, A-31573); Donkey anti-Goat Alexa Fluor 647 (1:500, Thermo Fisher Scientific, A-21447).

### Lineage tracing analysis

**Quantification of Lin$^{3a2}$ area as a percentage of Krt5$^+$ area.** The composite image of lung sections stained with Rabbit anti-Krt5(green; Cell Marque: EP1601Y), Gt anti-tdT (red; Scigen: Ab8181-200), and DAPI (blue) were imported into Fiji image analysis software as separate images based off their respective channels. Positive staining for Krt5 and tdT was determined using Fiji's 'Threshold' function: Image → adjust → threshold. A separate image showing overlapping pixels between the Krt5 and tdT image was generated using the 'AND' operation in Fiji's image calculator function: Process → Image calculator. Areas containing negatively stained nuclei (or 'holes') were resolved using Fiji's Watershed function: Process → Binary → Watershed. Krt5 and Krt5-tdT overlapping areas were measured using the following sequence of features: Analyze → Analyze particles. Percent tdT/Krt5 area was calculated by Krt5-tdT area as a function of the total Krt5 area.

**Quantification of cells per unit BM.** To score the number of cells per unit BM, 20 lines along the basement membrane were measured using the 'segmented line tool' in the Fiji image analysis software[73] and recorded into a Google spreadsheet. The number of lineage-tagged tdT and eGFP from DR mice was counted along a basement membrane length of 300 μm.

**Quantification of Pod region percent totals.** Pods were defined as at least five continuous Krt5/eGFP/tdT-immunoreactive cells in alveolar regions that were not associated with pre-existing bronchial epithelium. 100 pod cells per region of interest were counted using Fiji's 'multi-point' function and recorded into a Google spreadsheet. Then the percentage of either tdT low, tdT high, or eGFP positive cells was assessed based on the function of counted pod cells.

**Quantification of Krt5 area in *IL-22$^{Cre}$* and *IL-22ra1$^{fl/fl}$/Shh-Cre***
The composite image of lung sections stained with Ck anti-Krt5 (green; BioLegend: 905901), Hm anti-Pdpn (red; LifeSpan Biosciences: LS-C143022-100), and DAPI (blue) were imported into Fiji image analysis software as separate images based off their respective channels. Images were converted into greyscale using the following sequence of image processing features: Image → Type → 8 bit. Area's containing positive stains were then converted into binary black/white images using the following sequence of image processing features: Process → Binary → Make Binary. Area of each channel was measured and recorded into a Google spreadsheet using the following sequence of features: Analyze → Analyze particles. Quantification of Krt5 area was calculated by measuring the area of Krt5$^+$ immunoreactive pods in proportion to the area of damaged regions demarcated by DAPI$^+$PDPN$^-$ stain within alveolar epithelium using Fiji software.

### Quantitative real-time PCR
RNA was extracted from homogenized snap-frozen superior lobes of either *IL-22$^{Cre}$* homozygous, *IL-22ra1$^{fl/fl}$/Shh-Cre*, or WT controls using RNAeasy mini kit (Qiagen, cat: 74106). Isolated RNA was converted into cDNA using an iScript cDNA synthesis kit (Bio-Rad, cat: 1708891). Gene expression analysis was performed using SYBR Green PCR master Mix

(Thermo Fisher Scientific, cat: 4309155) and analyzed on the 7500 fast Real-Time PCR system (Thermo Fisher Scientific).

### Flow cytometry of IL-22-expressing cells
*IL-22$^{Cre}$* homozygous mice were purchased and bred to heterozygosity with another strain expressing at least one copy of *ROSA-tdT* to generate *IL-22$^{Cre}$/ROSA-tdT* mice. Mouse lungs were collected from a steady state and from mice infected with influenza for 14 days. Lungs were collected and processed into single-cell suspension as described above in the 'Preparation of single-cell suspensions for Single-cell RNAseq' section. An immune panel capable of delineating between T-helper and non-T-helper subsets (CD3$^+$CD4$^+$tdT$^+$ and CD3$^+$CD4$^-$tdT$^+$, respectively) was used to determine the main IL-22-expressing cells using flow cytometry.

### Multiplex protein assays
Mouse bronchial alveolar lavage and mouse left lobes were collected at the following time points: naïve, 3, 5, 7, 9, 11, 14, 17, 21 dpi. BAL was prepared through the instillation of intubated mouse trachea with 1 ml 1X dPBS three times. Instilled 1X dPBS was transferred into a 1.5 microfuge tube. To prepare lung homogenates, dissected lobes were transferred into collection tubes containing 1.4 mm ceramic beads (Lysing matrix D, MP Biomedicals Cat: 116913100) and homogenized using mechanical agitation (MP Benchtop Homogenizer, MP biomedicals, Cat: 6VFV9). Samples were centrifuged at 600×*g* to pellet cells. Supernatants were transferred into a separate 1.5 ml collection tube. A multiplexed protein assay (Bio-plex Pro, Bio-Rad, Cat: 171304070, M69999997NY) was performed to assess changes in the expression of the following cytokines: IL-1b, IL-6, IL-10, IL-17, IL-22, Tnf and Ifnγ. Samples were processed for analysis as per manufacturer instructions. Cytokine levels were quantified using the fluid flow-based microplate reader (Bio-plex 200, Luminex, Cat:171000201).

### Quantification and statistical analysis
Detailed descriptions relating to the quantification and statistical analysis of each experiment are documented in the above "Methods" section. Statistical analysis from graphs generated by wet lab experiments was performed in Graphpad Prism 7. Variance in datapoints between conditions is represented by mean ± SEM. Statistical analysis for single-cell RNA seq data was performed using pipelines for statistical analysis inherit to each package.

### Reporting summary
Further information on research design is available in the Nature Portfolio Reporting Summary linked to this article.

## Data availability
The data discussed in this publication have been deposited in NCBI's Gene Expression Omnibus[74] and are accessible through GEO Series accession number GSE184384. All other data are available in the article and its Supplementary files or from the corresponding author upon request. Source data are provided with this paper. A Figshare link is provided with this paper: (https://doi.org/10.6084/m9.figshare.23907696). Source data are provided with this paper.

## Code availability
URL link to Github for code: https://github.com/BeppuAN/20220912_GEO-asssession-GSE184384.

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

## Acknowledgements

We would like to thank Matt Kostelny and Guangzhu Zhang for assistance with animal husbandry, and Stephen Beil for general laboratory support. We acknowledge support from the Applied Genomics, Pavilions Flow Cytometry & Cell Sorting, and Mouse Genetics cores at Cedars Sinai Medical Center (CSMC). This research was supported by grants from the National Institutes of Health (NIH) (R01 HL135163; P01 HL108793) to B.R.S., by the Bram and Elain Goldsmith Chair in Gene Therapeutics Research, and by the Office of Graduate Education at CSMC.

## Author contributions

Conceptualization, A.K.B., B.R.S.; methodology, A.K.B., J.Z., C.Y., G.C., E.I., K.D., A.L.C., B.R.S.; data analysis, A.K.B., B.R.S.; investigation, A.K.B., J.Z., C.Y., K.D., B.R.S.; resources, E.I., A.L.C., C.M.H., J.K.K., B.R.S.; writing—original draft, A.K.B., B.R.S.; writing—review & editing, A.K.B., J.Z., C.Y., G.C., E.I., A.L.C., C.M.H., J.K.K., W.C.P., B.R.S.; visualization, A.K.B., B.R.S.; supervision, A.K.B., J.Z., C.Y., G.C., W.C.P., B.R.S.; and funding acquisition, B.R.S.

## Competing interests

The authors declare no competing interests.
