## [Peer Review File · Nature Communications]

Epithelial Plasticity and Innate Immune Activation Promote Lung Tissue Remodeling following Respiratory Viral InfectionREVIEWER COMMENTS

Reviewer #1 (Remarks to the Author):

General:

The authors argue that acute lung injury and chronic lung disease such as viral infections and IPF, disrupt normal progenitor cell compartmentalization leading to aberrant tissue remodeling and declining lung function. Their study addresses the identity of epithelial progenitor cells that contribute to proximalization of distal lung tissue and the mechanisms that regulate their fate during tissue remodeling. The authors convincingly show that intralobular serous cells (Lin3a2-high) are progenitors of the hyperplastic basal cells, and that IL-22 promotes the expansion of these basal cells. The methods and models and presentation of the data are of a high standard, and raises no questions. There are, however, a couple of issues that need some clarification.

Major comments:

1. Whereas this reviewer understands the argument for a disrupted progenitor compartmentalization in IPF, this seems less solid for respiratory viral infections. The manuscript would gain by having some references to studies showing (indications for) a disrupted progenitor compartmentalization in response to respiratory viral infections in humans. Or were the authors referring only to the respiratory viral infections that lead to acute lung injury. It is important to clarify this to readers.
2. There is considerable attention for the role of IL-22 in the expansion of the hyperplastic basal cells (hBC), which I do understand. This reviewer was intrigued also by the expansion of Ifitm1, Ifitm3, Bpifa1, and Ltf-expressing serous cells in response to the viral challenge. Apparently, a small population of these cells, expressing among others Bpifa1, are the progenitor cells of the hBC. There is little discussion, however, on e.g. the role of interferon in stimulating these serous cells. Do you consider this to be part of the innate antiviral response? Is there a relation between the amount of interferon and the generation of these progenitor cells. Was there a particular reason why you did not follow this up. This reviewer would appreciate some clarification and discussion.
3. Multiple additional processes have been implicated in halting/restoring acute lung injury (angiotensin, amphiregulin, migration of cells, etc) and secondary bacterial infections (CD200, IDO), but nothing really has been brought up in the discussion to try and put your

findings in the context of what is known. I do fully appreciate that this is not a review, but some context is highly informative to readers.

Minor comment:

1. At line 258/9 you refer to increased postnatal susceptibility to allergic inflammation. Apart from the fact that I believe there is no consensus on this and the statement is somewhat beyond the lead of this manuscript, after re-reading Hackett's paper I failed to find evidence for that, or did I miss it.

R. Lutter

Reviewer #2 (Remarks to the Author):

The manuscript by Andrew K. Beppu et al. reports a novel aspect of epithelial plasticity in lungs following influenza virus infection. They show that intralobar serous (IS) cells can assume basal cell fates, whereby those basal cells require IL22 to colonize injured alveoli. However, ultimately these cells fail to replace normal alveolar epithelium and re-differentiate to IS, resulting in distal lung remodeling.

This manuscript is extremely well written and offers interesting and novel insights into lung regeneration. This study is important, especially because the renewal and repair of injured distal, alveolar epithelia following respiratory viral infections is poorly understood.

Considering the current interest in respiratory infections and long term damage to lung tissues, this study will be without doubt of interest to the wider scientific community.

Major Comments:

1. The potential impact of impaired/altered epithelial regeneration/repair in the absence of IL22 is interesting and it would be exciting to better understand the biological, long-term consequence. The timepoints after PR8 infection investigated in this study vary between experiments and this slight inconsistency makes it difficult to appreciate the overall effect. For instance, 5-11dpi (e.g. line 268) is called "recovery phase", which is incorrect. Infection with PR8 results in the increasing attraction of (first) innate, then adaptive immune cells to the site of infection. Peak inflammation is approximately after 9 days, which is also visible by max. weight loss and inflammation associated tissue damage. Only after this time does the

recovery start – which is usually associated with weight gain and decrease in lung infiltrates. It seems that early expansion of BC (i.e. before 14dpi) occurred independent of IL22, whereas IL22 or its receptor was clearly required at 14dpi (Fig. 6C and 6F). One would assume that the potentially detrimental consequences of this impaired BC proliferation might result in impaired healing – which would then be visible at even later stages. Was there any difference in weight loss, or re-gaining weight later than 14dpi (Suppl 6B only shows 14dpi)? Was there a difference in the overall damage of lung tissue? If I understand e.g. Suppl Fig. 6C correctly, there seems to be more inflammatory infiltrates (i.e. Pdpn negative, Dapi+ areas) in those lung.

Please explain, make sure consistent terminology is used when referring to the e.g. early inflammatory phase, peak inflammation, recovery, resolution and it might be helpful to provide conventional H&E stains in the Suppl file to comprehend the long term effects of altered IL22 mediated cell repair processes.

2. I believe better (or additional) references should be chosen when discussing the role of innate immunity in epithelial cell repair. I provide 2 examples below:

Line 257: claim innate immune activation in response to infection is a key regulator of epithelial cell fate leading to remodeling and increased postnatal susceptibility to allergic inflammation (Hackett 2011). The reference uses in vitro cultures of airway cells from asthmatic and non-asthmatic people, exposes them to RSV or environmental air and say that asthmatic cells make more IL6. That is not what the authors say.

Line 259: Notably, local production of TNF and IL- 1 following influenza virus infection in mice promote the regenerative capacity of alveolar epithelium (Katsura et al., 2019). This paper uses airway organoids (screening assay) and describe that those 2 cytokines act on AEC2 cells, suggesting a role in AEC2 mediated lung regeneration - they also do in vivo PR8 and show that in IL1R-Ko mice less cells proliferate – still, this is no clear way to show a role of immune-cell mediated effects (organoids is main thing, and starting point, i.e. lacking immune cells), and the in vivo work only shows that CD45+ cells are there.

Please correct, or replace, or amend.

3. I have one question, just for clarification: (line 368) can it be that in the absence of IL22 BC do not develop from IS cells? i.e. the defect in the absence of IL22 is not a premature

dedifferentiation BC>IS, but the lack of differentiation (IS>BC)? Or is this considered impossible because only hBC express IL22R?

Minor comments:

Line 609, Figure 1A: are those cells GFP+ (according to Figure legend) or GFP- (according to Figure), or tdT- as stated in the methods section? Considering Sftpc-CreER mice received TM, I assume you sorted for tdT+ and GFP- cells? Please clarify.

Line 196: should refer to Fig. 3C (not D)

Line 319: should be Fig 6C, D

Reviewer #3 (Remarks to the Author):

The manuscript by Beppu and colleagues entitled “Epithelial Plasticity and Innate Immune Activation Promote Lung Tissue Remodeling following Respiratory Viral Infection” explores the role of infection and injury in lung plasticity and remodeling. In particular they report that intralobar serous cells (IS) in the distal airway are activated to produce hyperplastic basal cells (“proximalization” of distal lung) with resulting increases in antimicrobial defense, but eventual loss of lung function.

The study utilizes the well established PR8 influenza model to study lung tissue remodeling post-infection using more sophisticated single cell and lineage studies than previously have been identified. It includes comprehensive single cell transcriptomics and multiple time points after PR8 infection, spatial RNA-seq, and an impressive set of GEMMs including an innovative dual reporter system that facilitated lineage studies particularly of IS and Club cells. While prior studies have identified a role of IL-22 in this process, this study details the specific populations involved in the process in more detail and reports that a novel population of distal airway secretory cells (intralobar serous or IS cells) are activated to assume an hBC fate, contributing to the remodeling in distant airways. They also provide greater insights into the immune populations modulated by IL-22. Overall it makes

significant contributions to our understanding of epithelial plasticity and lung remodeling after viral infection, a process that may have broader implications for lung syndromes after other infections (e.g. Covid) and fibrotic processes in the lung. There are a few points that should be addressed.

1. It would be useful to further discuss the findings of this study in the context of the earlier study by Pociask et al (PMID: 23490254) referenced in the text studying the impact of IL-22 knockout on lung repair after PR8 influenza infection. In that study they reported that the IL22 knockout mice had exacerbated lung injury with decreased lung function 21 days post-infection, as well as a fibrotic phenotype, compared with wt mice. In the present study, they remark that epithelial plasticity (driven by IL-22) protects against mortality from acute respiratory viral infection but results in distal lung remodeling and loss of lung function. There seems to be a discrepancy between the studies in terms of the ultimate impact of IL-22 on lung function.

A. Do the authors propose that IL-22 driven hBC proliferation and the subsequent remodeling, with “proximalization” of distant airways, reduces lung function as stated at the end of the abstract? It does not seem there is direct functional evidence of this, but rather just phenotypic changes in the composition of the distal airways, in which case the impact on lung function may not be established.

B. The authors state “These data shed new light on mechanisms of fibrosis in lungs of IL-22^{-/-} mice following PR8 infection (Pociask et al., 2013) where IL-22 restrains hBC in a highly proliferative and migratory state allowing expansion and reepithelialization of injured airways and alveoli”. Please clarify how this sheds light on the mechanism of fibrosis- do they believe the fibrosis results from a lack of hBC and hence reduced re-epithelialization? Alternatively, do these hBCs or other stromal cells differentiate towards a more mesenchymal (and likely pro-fibrotic) state?

C. The authors should discuss whether there are potential therapeutic implications of the mechanism they propose- does it support activating the IL-22 pathway after viral infection or other injury to reduce superinfection or the fibrotic reaction (but potentially reducing lung function due to remodeling according to the authors), or blocking the IL-22 pathway to prevent the aforementioned loss of lung function? It is a complex mechanism but clarifying the potential implications for the reader would be useful.

Minor point: could the authors clarify why there are tdT bright (Lin3a2-high) and tdT-dim (Lin3a2-low) populations? It is unclear from dual reporter system why this should occur or be related to lineage (correlating with Krt5- and Krt5+immunoreactivity, respectively)

Reviewer #4 (Remarks to the Author):

The comments are primary on data analysis for single cell and spatial gene expression data.

The overall scRNASeq analysis processes are reasonable, but some details should be clarified without checking for the code.

1. The author should present a UMAP of all cells (not just epithelial cells in Figure 1B) to give better idea of overall cell population.
2. The single cell data are from multiple time points, it would be great to have color coded time point dimensional plots for all cells and epithelial cells.
3. More detail about the epithelial cell type annotation, by checking the code, the marker genes are used, the author should give some references for those markers.
4. More details are needed for subsetting data in line 967 and 968.
5. The author used Velocity and scVelo to study the cell trajectory, pseudotime analysis could be very helpful given that the data are gathered at different data point.
6. In Figure 2F, are spatial data and the immunofluorescence data from same exact position? They do not match well.
7. In Figure 3C, lower row, D, E F, could the author do the colonization calculation other than the representative figures?
8. Figure 4D, a better color schema should be used instead of the Seurat default. It is hard to separate a few cell populations. Also a UMAP colored by the sample would be helpful to check the relationship on immune cells.
9. Supposedly Figure 2E and Supp. Figure 2B are from the same data, which one is skewed? Please keep the aspect ratio as 1 all spatial plots.

**RESPONSE TO REVIEWER COMMENTS:**

We would like to thank reviewers for their constructive suggestions for improvement of our manuscript.
Specific responses to questions and concerns are included below for each reviewer:

**Reviewer #1 (Remarks to the Author):**

General:

The authors argue that acute lung injury and chronic lung disease such as viral infections and IPF, disrupt
normal progenitor cell compartmentalization leading to aberrant tissue remodeling and declining lung
function. Their study addresses the identity of epithelial progenitor cells that contribute to proximalization of
distal lung tissue and the mechanisms that regulate their fate during tissue remodeling. The authors
convincingly show that intralobular serous cells (Lin3a2-high) are progenitors of the hyperplastic basal cells,
and that IL-22 promotes the expansion of these basal cells. The methods and models and presentation of the
data are of a high standard, and raises no questions. There are, however, a couple of issues that need some
clarification.

Major comments:

1. Whereas this reviewer understands the argument for a disrupted progenitor compartmentalization in IPF,
this seems less solid for respiratory viral infections. The manuscript would gain by having some references to
studies showing (indications for) a disrupted progenitor compartmentalization in response to respiratory viral
infections in humans. Or were the authors referring only to the respiratory viral infections that lead to acute
lung injury. It is important to clarify this to readers.

Thank you for this suggestion. We include additional references in the revised manuscript that document
proximalization of parenchymal tissue of the injured human lung in the setting of diffuse alveolar damage
secondary to bacterial pneumonia (Taylor et al., 2018), Influenza A virus infection (Keeler et al., 2018), and
SARS-CoV2 (Wu et al., 2023), Page: 3; Line: 49 .

2. There is considerable attention for the role of IL-22 in the expansion of the hyperplastic basal cells (hBC),
which I do understand. This reviewer was intrigued also by the expansion of Ifitm1, Ifitm3, Bpifa1, and Ltf-
expressing serous cells in response to the viral challenge. Apparently, a small population of these cells,
expressing among others Bpifa1, are the progenitor cells of the hBC. There is little discussion, however, on e.g.
the role of interferon in stimulating these serous cells. Do you consider this to be part of the innate antiviral
response? Is there a relation between the amount of interferon and the generation of these progenitor cells.
Was there a particular reason why you did not follow this up. This reviewer would appreciate some
clarification and discussion.

This is an interesting point raised by the reviewer. Even though IFN signaling does not represent a major focus
of our study, we have included additional discussion in the revised manuscript due to the likely coordinated
regulation of Ifitm isoforms by IAV-elicited IFN signaling within IS cells (Supp. Fig. 1C). Previous studies have
shown that this may be mediated by a common IFN-responsive enhancer (Li et al., 2017).

The reviewer also asks if there is a relationship between interferon signaling and regulation of IS>BC
progenitors. Although we provide evidence of increased interferon signaling among IS cells, gene ontology
terms upregulated among hBC's included cytokine signaling networks but did not show a prominent IFN

signaling signature (Figure 4A). Accordingly, we focused on cytokine signaling, particularly on roles for IL-22, in
regulating serous-basal-serous differentiation. However, we acknowledge potential roles for IFN signaling and
we have revised the text to reiterate this point as follows:

Page: 5; Line: 135; 'Similarly, other antimicrobial genes such as Bpifa1 and Ltf, whose expression defines
serous cells of proximal airways and submucosal glands (Tata et al., 2018), were both induced following PR8
infection and served to distinguish the transcriptomes of serous cells from the other bronchiolar secretory cell
type, club cells (Fig. 1F; Supp. Fig. 2A)'.

3. Multiple additional processes have been implicated in halting/restoring acute lung injury (angiotensin,
amphiregulin, migration of cells, etc) and secondary bacterial infections (CD200, IDO), but nothing really has
been brought up in the discussion to try and put your findings in the context of what is known. I do fully
appreciate that this is not a review, but some context is highly informative to readers.

We have included additional discussion to link acute lung injury and secondary bacterial infections with
cellular and transcriptomic changes among epithelial cell types. However, we respectfully feel that extending
this to a broader discussion of lung injury that is secondary to severe respiratory viral infection is beyond the
focus of this study.

Minor comment:

1. At line 258/9 you refer to increased postnatal susceptibility to allergic inflammation. Apart from the fact
that I believe there is no consensus on this and the statement is somewhat beyond the lead of this manuscript,
after re-reading Hackett's paper I failed to find evidence for that, or did I miss it.

Thank you for this comment and we agree. We have removed discussion of this study from the revised
manuscript to avoid confusion.

**Reviewer #2 (Remarks to the Author):** 72

The manuscript by Andrew K. Beppu et al. reports a novel aspect of epithelial plasticity in lungs following
influenza virus infection. They show that intralobar serous (IS) cells can assume basal cell fates, whereby those
basal cells require IL22 to colonize injured alveoli. However, ultimately these cells fail to replace normal
alveolar epithelium and re-differentiate to IS, resulting in distal lung remodeling.

This manuscript is extremely well written and offers interesting and novel insights into lung regeneration. This
study is important, especially because the renewal and repair of injured distal, alveolar epithelia following
respiratory viral infections is poorly understood. Considering the current interest in respiratory infections and
long term damage to lung tissues, this study will be without doubt of interest to the wider scientific
community.

Major Comments:

1. The potential impact of impaired/altered epithelial regeneration/repair in the absence of IL22 interesting
and it would be exciting to better understand the biological, long-term consequence. The timepoints after PR8
infection investigated in this study vary between experiments and this slight inconsistency makes it difficult to
appreciate the overall effect. For instance, 5-11dpi (e.g. line 268) is called "recovery phase", which is incorrect.
Infection with PR8 results in the increasing attraction of (first) innate, then adaptive immune cells to the site of
infection. Peak inflammation is approximately after 9 days, which is also visible by max. weight loss and

inflammation associated tissue damage. Only after this time does the recovery start – which is usually
associated with weight gain and decrease in lung infiltrates.

The reviewer suggests changing confusing terminology such as ‘recovery phase’ as it has a specific definition
in the context of the innate and adaptive immune response. Thank you for this suggestion, and we agree with
the reviewer. This has been change in the revised manuscript, simply referring to time post initial infection by
PR8.

2. It seems that early expansion of BC (i.e. before 14dpi) occurred independent of IL22, whereas IL22 or its
receptor was clearly required at 14dpi (Fig. 6C and 6F). One would assume that the potentially detrimental
consequences of this impaired BC proliferation might result in impaired healing – which would then be visible
at even later stages. Was there any difference in weight loss, or re-gaining weight later than 14dpi (Suppl 6B
only shows 14dpi)? Was there a difference in the overall damage of lung tissue? If I understand e.g. Suppl Fig.
6C correctly, there seems to be more inflammatory infiltrates (i.e. Pdpn negative, Dapi+ areas) in those lung.
Please explain, make sure consistent terminology is used when referring to the e.g. early inflammatory phase,
peak inflammation, recovery, resolution and it might be helpful to provide conventional H&E stains in the
Suppl file to comprehend the long term effects of altered IL22 mediated cell repair processes.

The reviewer suggests additional experiments investigating body weight changes and conventional H&E
staining at later timepoints to determine roles for reduced BC-hyperlasia and its impact on healing in IL-22 LOF
mice. We agree with the reviewer that determining the role of hBC in regard to their contribution to survival is
of importance. Related to this issue, Pociask et al. documented body weight changes and histological changes
by H&E staining at later timepoints (21 days post-infection) in PR8 infected IL-22 KO mice (Pociask et al., 2013,
PMID: 23490254). These experiments were not repeated in the present study. Instead we have focused on the
events associated with altered BC abundance, proliferation and differentiation during the early regenerative
phase following PR8 infection. Further studies aimed at defining the long-term consequences of altered IL22
signalling that are beyond those studies detailed in the Pociask et al. study would be of interest, but we
respectfully feel that this represents an area for future investigation that is beyond the scope of the present
study.

Minor comments:

Line 609, Figure 1A: are those cells GFP+ (according to Figure legend) or GFP- (according to Figure), or tdT- as
stated in the methods section? Considering Sftpc-CreER mice received TM, I assume you sorted for tdT+ and
GFP- cells? Please clarify.

Thank you for pointing this out and we have revised the figure legend accordingly. AT2 cells were lineage
labelled as indicated, identified based upon their GFP fluorescence, and GFP- epithelial cells enriched by FACS.
The following additional changes have been made to the revised manuscript:

Page: 19; Line: 673 - ‘either total epithelial cells (C57/Bl6 mice;CD31-CD45-CD326+) or AT2...’

Page: 19; Line: 674 - ‘(Sftpc-CreER/ROSA-mTmG mice; CD31-CD45-CD326+eGFP-) enriched by FACS...’

Page:40: line: 008 - ‘negative epithelium (i.e. CD326+CD45-CD31-eGFP-) by FACS. For...’

Line 196: should refer to Fig. 3C (not D)

Thank you for pointing this out – this has been corrected in the revised manuscript (Page: 7; Line: 204 -
‘immunofluorescent staining for Bpifa1, and were uniformly negative for markers of ciliated and BC (Fig. 3C).’)

Line 319: should be Fig 6C, D

This has been changed in the revised manuscript to simply refer to Fig. 6D. (Page: 9; Line: 332 - 'which reached
statistical significance by the day 17 post-infection time point (Fig. 6D)')

**Reviewer #3 (Remarks to the Author):**

The manuscript by Beppu and colleagues entitled "Epithelial Plasticity and Innate Immune Activation Promote
Lung Tissue Remodeling following Respiratory Viral Infection" explores the role of infection and injury in lung
plasticity and remodeling. In particular they report that intralobar serous cells (IS) in the distal airway are
activated to produce hyperplastic basal cells ("proximalization" of distal lung) with resulting increases in
antimicrobial defense, but eventual loss of lung function.

The study utilizes the well established PR8 influenza model to study lung tissue remodeling post-infection
using more sophisticated single cell and lineage studies than previously have been identified. It includes
comprehensive single cell transcriptomics and multiple time points after PR8 infection, spatial RNA-seq, and
an impressive set of GEMMs including an innovative dual reporter system that facilitated lineage studies
particularly of IS and Club cells. While prior studies have identified a role of IL-22 in this process, this study
details the specific populations involved in the process in more detail and reports that a novel population of
distal airway secretory cells (intralobar serous or IS cells) are activated to assume an hBC fate, contributing to
the remodeling in distant airways. They also provide greater insights into the immune populations modulated
by IL-22. Overall it makes significant contributions to our understanding of epithelial plasticity and lung
remodeling after viral infection, a process that may have broader implications for lung syndromes after other
infections (e.g. Covid) and fibrotic processes in the lung. There are a few points that should be addressed.

1. It would be useful to further discuss the findings of this study in the context of the earlier study by Pociask
et al (PMID: 23490254) referenced in the text studying the impact of IL-22 knockout on lung repair after PR8
influenza infection. In that study they reported that the IL22 knockout mice had exacerbated lung injury with
decreased lung function 21 days post-infection, as well as a fibrotic phenotype, compared with wt mice. In the
present study, they remark that epithelial plasticity (driven by IL-22) protects against mortality from acute
respiratory viral infection but results in distal lung remodeling and loss of lung function. There seems to be a
discrepancy between the studies in terms of the ultimate impact of IL-22 on lung function.

We have included additional discussion relating findings of this study to those of Pociask, as suggested by the
reviewer. We have also revised text Page: 2; Line: 31 as follows: "However, re-epithelialization of injured
alveoli by BC and their differentiated progeny increases expression of antimicrobial factors that have potential
to protect against secondary infection and associated morbidity/mortality, but fails to restore normal alveolar
epithelium responsible for gas exchange."

173 A. Do the authors propose that IL-22 driven hBC proliferation and the subsequent remodeling, with
174 "proximalization" of distant airways, reduces lung function as stated at the end of the abstract? It does not
seem there is direct functional evidence of this, but rather just phenotypic changes in the composition of the
distal airways, in which case the impact on lung function may not be established.

We agree with the reviewer and have removed claims of functional changes and have simply stated the
structural changes to lungs of mice and infer from this the potential for functional changes (see above).

B. The authors state “These data shed new light on mechanisms of fibrosis in lungs of IL-22^{-/-} mice following
PR8 infection (Pociask et al., 2013) where IL-22 restrains hBC in a highly proliferative and migratory state
allowing expansion and reepithelialization of injured airways and alveoli”. Please clarify how this sheds light on
the mechanism of fibrosis- do they believe the fibrosis results from a lack of hBC and hence reduced re-
epithelialization? Alternatively, do these hBCs or other stromal cells differentiate towards a more
mesenchymal (and likely pro-fibrotic) state?

We have revised text in the manuscript to emphasize that these data shed new light on potential mechanisms
and include additional discussion as to mechanism – i.e. either the ability of hBC’s to re-epithelialize denuded
basement membrane and thus suppress fibroproliferative responses. (Page: 12; Line: 450 - ‘Through a
combination of IS>BC>IS phenotypic plasticity, we posit that the suppressive effect of IL-22 signalling on lung
fibroproliferative responses to PR8 infection results from its influence on BC expansion and arrested
differentiation.’)

C. The authors should discuss whether there are potential therapeutic implications of the mechanism they
propose- does it support activating the IL-22 pathway after viral infection or other injury to reduce
superinfection or the fibrotic reaction (but potentially reducing lung function due to remodeling according to
the authors), or blocking the IL-22 pathway to prevent the aforementioned loss of lung function? It is a
complex mechanism but clarifying the potential implications for the reader would be useful.

Thank you for this suggestion and we have included additional discussion of the potential therapeutic
implications of this work in the discussion section of the manuscript. (Page: 12; Line: 459 - ‘Interestingly, in
other prior studies, IL-22 has been shown to protect against influenza virus-induced pneumonia (Hebert et al.,
2020) and has been implicated in the production of antimicrobial factors that protect against secondary
bacterial infection following influenza virus infection (Abood et al., 2019).’)

Minor point: could the authors clarify why there are tdT bright (Lin3a2-high) and tdT-dim (Lin3a2-low)
populations? It is unclear from dual reporter system why this should occur or be related to lineage (correlating
with Krt5- and Krt5+immunoreactivity, respectively)

This is a good question. It is clear from others and us in the published literature that distinct clonal populations
harbouring ROSA26-based lineage tracers can express reporters at different levels (Snippert et al.,
PMID: 20887898 and McConnell et. al., PMID: 27880895). Cell type and tissue-specific differences in CreER
activity have also been observed between tissues and cell types
(<https://www.informatics.jax.org/allele/MGI:2182767?recomRibbon=open>). We believe that the observed
differences in RFP fluorescence intensity relate more to the latter, where the absolute level of reporter
expression from the recombined ROSA26 allele varies between cell types. These findings are further
supported by results from organoid cultures, where clonally-derived organoids generated from RFP+ IS cells
contain both Lin3a2-high and Lin3a2-low cell types (Figure 3K). Another possibility that we have not ruled out
is that differences in fluorescence intensity may be related to the subcellular distribution of the fluorescent
protein, where the predominantly cytoplasmic distribution of RFP represents a much larger pool of fluorescent
protein in IS cells compared to BC.

**Reviewer #4 (Remarks to the Author):**

The comments are primary on data analysis for single cell and spatial gene expression data.

The overall scRNASeq analysis processes are reasonable, but some details should be clarified without checking
for the code.

For the benefit of this reviewer, scRNAseq data can be accessed using GEO token # qzozsekedhoxrof.

Responses to other questions/concerns include:

1. The author should present a UMAP of all cells (not just epithelial cells in Figure 1B) to give better idea of
overall cell population.

Single cell capture, library preparation and RNAseq were performed using independently fractionated
epithelial and immune cells only. As such, UMAP projections display all cells captured for each of these lung
cell fractions.

2. The single cell data are from multiple time points, it would be great to have color coded time point
dimensional plots for all cells and epithelial cells.

We have provided UMAP projections within supplemental data (Supplemental Figures 1A and 5) that define
epithelial and immune cells sampled for each of the time points included in the combined dataset.

3. More detail about the epithelial cell type annotation, by checking the code, the marker genes are used, the
author should give some references for those markers.

References to markers shown in Supp Figure 1A include the following, that have been included in the Methods
section of the revised manuscript (Page: 41; Line: 035): "Cell type annotation was performed using the
following cell type-specific markers: Ltf (Serous cells; (Deprez et al., 2020; Raphael et al., 1989)), Krt5, Trp63
(Basal cells; (Daniely et al., 2004; Schoch et al., 2004)), Cldn10, Scgb1a1 (Club cells; (Singh et al., 1990; Zemke
et al., 2009)), Ager (Alveolar Type I cells; (Chung and Hogan, 2018)), Sftpc, Chil1, Hc, Scid1 (Alveolar Type II
cells; (Du et al., 2015)), Foxj1 (Ciliated cells; (You et al., 2004))."

4. More details are needed for subsetting data in line 967 and 968.

Thank you for this suggestion. Additional detail has been provided in the revised manuscript. (Page: 41; Line:
028 - 'The R package 'Seurat' (Butler et al., 2018; Stuart et al., 2019) was used to apply standard quality
control metrics and unsupervised clustering (FindClusters(x, resolution = 0.5) in Seurat) for generation of initial
UMAP projections. Cell types were annotated based on gene expression for canonical markers and saved into
object metadata (object@meta.data\$anno in Seurat).')

5. The author used Velocity and scVelo to study the cell trajectory, pseudotime analysis could be very helpful
given that the data are gathered at different data point.

The reviewer raises a valid point regarding use of other algorithms to assess differentiation trajectory.

However, we elected to validate observations made using Velocity through lineage tracing and we
respectfully feel that further bioinformatic analysis of differentiation trajectories would be redundant.

6. In Figure 2F, are spatial data and the immunofluorescence data from same exact position? They do not
match well.

No, visium RNAseq and immunofluorescence analysis were performed on separate tissue sections. We have
revised the figure legend to clearly make this point.

7. In Figure 3C, lower row, D, E F, could the author do the colonization calculation other than the
representative figures?

Quantification of Co-localization for Fig. 3C, D, E and F are included in Supp. Fig. 3H, and I.

8. Figure 4D, a better color schema should be used instead of the Seurat default. It is hard to separate a few
cell populations. Also a UMAP colored by the sample would be helpful to check the relationship on immune
cells.

Thank you for the suggestion, Color schema for the immune single cell dataset has been changed. An
additional supplemental figure has been generated to track changes in immune cell populations as a function
of time following PR8 exposure.

9. Supposedly Figure 2E and Supp. Figure 2B are from the same data, which one is skewed? Please keep the
aspect ratio as 1 all spatial plots.

Thank you for the suggestion. Aspect ratio for various panels mentioned have been aligned.

REVIEWERS' COMMENTS

Reviewer #1 (Remarks to the Author):

I like to thank the authors for their responses and changes. In all I do feel this improved the - already quite good - manuscript. There are just some minor issues.

Minor comments:

1. The sentence added to the abstract does repeat some aspects from the previous sentence, which is not needed to make their point.
2. Suppl. fig 1C is mentioned in the text after suppl. figure 1D?
3. Line 419 should read number and in line 473, associated is used twice.

Rene Lutter

Reviewer #2 (Remarks to the Author):

The authors have addressed all my concerns.

Reviewer #3 (Remarks to the Author):

The revised manuscript by Beppu and colleagues adequately addresses the issues raised in my prior review.

One minor point: in the discussion of the therapeutic implications of targeting IL-22 at the end of the Discussion, the authors restate what they expect would happen based on their model (that "pharmacologic targeting of IL-22 signaling may suppress BC expansion and associated parenchymal remodeling in lung diseases associated with proximalization of injured alveolar epithelium") but don't give the broader perspective as to the downsides (e.g. would also make someone more vulnerable to infections) and, in the balance, is the potentially worth doing. Does the upside of preventing remodeling outweigh potential downsides?

Reviewer #4 (Remarks to the Author):

Most of the concerns have been addressed and the analyses for the data are in good stand now.

Reviewer #1 (Remarks to the Author):

I like to thank the authors for their responses and changes. In all I do feel this improved the - already quite good - manuscript. There are just some minor issues.

Minor comments:

1. The sentence added to the abstract does repeat some aspects from the previous sentence, which is not needed to make their point.

Thank you for pointing this out. We have revised the abstract accordingly.

2. Suppl. fig 1C is mentioned in the text after suppl. figure 1D?

This has been corrected by revising Supplemental Figure 1 – panels C and D were switched and text in manuscript now cites panel C before panel D. D

3. Line 419 should read number and in line 473, associated is used twice.

This has been corrected.

Rene Lutter

Reviewer #2 (Remarks to the Author):

The authors have addressed all my concerns.

Reviewer #3 (Remarks to the Author):

The revised manuscript by Beppu and colleagues adequately addresses the issues raised in my prior review.

One minor point: in the discussion of the therapeutic implications of targeting IL-22 at the end of the Discussion, the authors restate what they expect would happen based on their model (that "pharmacologic targeting of IL-22 signaling may suppress BC expansion and associated parenchymal remodeling in lung diseases associated with proximalization of injured alveolar epithelium") but don't give the broader perspective as to the downsides (e.g. would also make someone more vulnerable to infections) and, in the balance, is the potentially worth doing. Does the upside of preventing remodeling outweigh potential downsides?

This has now been addressed in the revised discussion.

Reviewer #4 (Remarks to the Author):

Most of the concerns have been addressed and the analyses for the data are in good stand now.